# Non-invasive single-cell morphometry in living bacterial biofilms

Mingxing Zhang [1,4], Ji Zhang[1,4], Yibo Wang[1,4], Jie Wang [2], Alecia M. Achimovich[3], Scott T. Acton[2] & Andreas Gahlmann [1,3 ✉]

Fluorescence microscopy enables spatial and temporal measurements of live cells and cellular communities. However, this potential has not yet been fully realized for investigations of individual cell behaviors and phenotypic changes in dense, three-dimensional (3D) bacterial biofilms. Accurate cell detection and cellular shape measurement in densely packed biofilms are challenging because of the limited resolution and low signal to background ratios (SBRs) in fluorescence microscopy images. In this work, we present Bacterial Cell Morphometry 3D (*BCM3D*), an image analysis workflow that combines deep learning with mathematical image analysis to accurately segment and classify single bacterial cells in 3D fluorescence images. In *BCM3D*, deep convolutional neural networks (CNNs) are trained using simulated biofilm images with experimentally realistic SBRs, cell densities, labeling methods, and cell shapes. We systematically evaluate the segmentation accuracy of *BCM3D* using both simulated and experimental images. Compared to state-of-the-art bacterial cell segmentation approaches, *BCM3D* consistently achieves higher segmentation accuracy and further enables automated morphometric cell classifications in multi-population biofilms.

[1] Department of Chemistry, University of Virginia, Charlottesville, VA, USA. [2] Department of Electrical & Computer Engineering, University of Virginia, Charlottesville, VA, USA. [3] Department of Molecular Physiology & Biological Physics, University of Virginia School of Medicine, Charlottesville, VA, USA. [4] These authors contributed equally: Mingxing Zhang, Ji Zhang, Yibo Wang. ✉email: agahlmann@virginia.edu

Biofilms are multicellular communities of microorganisms that grow on biotic or abiotic surfaces[1–4]. In addition to cellular biomass, biofilms also contain an extracellular matrix (ECM), which is composed of polysaccharides, DNA, and proteins. Individual cells in biofilms interact with other cells, the ECM, or with the substrate surface, and the sum total of these interactions provide bacterial biofilms with emergent functional capabilities beyond those of individual cells. For example, biofilms are orders of magnitude more tolerant toward physical, chemical, and biological stressors, including antibiotic treatments and immune system clearance[1,2,5–8]. Understanding how such capabilities emerge from the coordination of individual cell behaviors requires imaging technologies capable of resolving and simultaneous tracking of individual bacterial cells in 3D biofilms.

Live-cell-compatible imaging technologies, such as optical microscopy, can reveal the spatial and temporal context that affects cellular behaviors. However, conventional imaging modalities are not able to resolve individual cells within thick 3D biofilms over extended periods of time. For example, the diffraction-limited lateral $x,y$-resolution (~230 nm) of a confocal fluorescence microscope is barely sufficient to resolve bacterial cells positioned next to each other on flat glass coverslips. Even worse, the diffraction-limited axial $z$-resolution (570 nm) is comparable to the size of a single bacterial cell, so that densely packed cells become unresolvable in the axial $z$-dimension[9,10]. Notable exceptions include loose biofilms (low cell density), spherical cell shapes[11,12], and mutant *Vibrio cholera* biofilms, in which cell–cell spacing is increased through the overproduction of ECM materials[13–15]. While single-cell-resolved images have been obtained in such special situations, conventional optical microscopy modalities are not generally capable to accurately resolve and quantitatively track individual cells in dense 3D biofilms.

While super-resolution derivatives of confocal microscopy, known as image scanning microscopy[16], can improve spatial resolution, a perhaps more important limitation for long-term live-cell imaging is photodamage to the specimen (phototoxicity) and to the fluorophores used for labeling (photobleaching)[17–19]. In confocal microscopy-based approaches, undesired out-of-focus fluorescence emission is filtered out by confocal pinholes to yield optically sectioned images with high contrast, i.e., high signal-to-background ratios (SBRs). However, repeated illumination of out-of-focus regions during laser scanning and high light intensities at the focal volume result in rapid photobleaching of fluorophores and unacceptable phototoxicity for light sensitive specimens[17–19]. In fact, confocal fluorescence microscopy (as well as its super-resolution derivatives) uses illumination light intensities that are two to three orders of magnitude higher than the light intensities under which life has evolved[18]. The high rates of phototoxicity and photobleaching make confocal-based microscopy unsuitable for high frame-rate time-lapse imaging of living specimens over many hours and days[14,15,17,20,21].

In recent years, light sheet-based fluorescence excitation and imaging approaches have been developed to overcome the drawbacks of confocal microscopy. Among these, lattice light sheet microscopy (LLSM)[18,19] and field synthesis variants thereof[22], axially swept light sheet microscopy[23,24], dual-view light sheet microscopy[25,26], and single-objective oblique plane light sheet microscopy[27–31] now combine excellent 3D spatial resolution with fast temporal resolution and low phototoxicity at levels that cannot be matched by confocal microscopy. Specifically, light sheet-based microscopy approaches can operate at illumination intensities that are below the levels of cellular phototoxicity, even for notoriously light sensitive specimens, and reduce fluorophore photobleaching by 20–50 times compared to confocal microscopy, while maintaining comparable spatial resolution and contrast/SBR[18,28].

An additional challenge in high-resolution biofilm imaging is data quantification. Even if sufficient resolution and high SBRs can be achieved to visually discern, i.e., qualitatively resolve individual cells, robust computational algorithms are still needed for automated cell segmentation and quantitative cell tracking. Toward this goal, image processing approaches based on the watershed technique and intensity thresholding have been developed over the years for single-cell segmentation in bacterial biofilms[13–15,21]. The broad applicability of watershed- and threshold-based image processing algorithms is however limited, because these algorithms require manual optimization of many user-selected parameters. Even with optimal parameters, watershed- and threshold-based image processing methods often produce suboptimal segmentation results, especially when cell densities are high, when SBRs are low, and when cellular fluorescence intensities are not uniform across the cytosol or the cell surface. To overcome the drawbacks of traditional mathematical image processing approaches, automated solutions based on supervised training of deep convolutional neural networks (CNNs) have been used in recent years with great success for a wide range of problems in biomedical image analysis[32].

Here, we present Bacterial Cell Morphometry 3D (*BCM3D*)[33], a generally applicable workflow for single-cell segmentation and shape determination in high-resolution 3D images of bacterial biofilms. *BCM3D* uses CNNs, in silico-trained with computationally simulated biofilm images, in combination with mathematical image analysis to achieve accurate single-cell segmentation in 3D. The CNNs employed in *BCM3D* are based on the 3D U-Net architecture and training strategy, which has achieved excellent performance in biomedical data analysis benchmark tests[32]. The mathematical image analysis modules of *BCM3D* enable post-processing of the CNN results to further improve the segmentation accuracy. We establish that experimental bacterial biofilms images, acquired by LLSM, can be successfully segmented using CNNs trained with computationally simulated biofilm images, for which the ground truth voxel-level annotation maps are known accurately and precisely. By systematically evaluating the performance of *BCM3D* for a range of SBRs, cell densities, and cell shapes, we find that voxel-level segmentation accuracies of >80%, as well as cell counting accuracies of >90%, can be robustly achieved. *BCM3D* consistently outperforms previously reported image segmentation approaches that rely exclusively on conventional image processing approaches. *BCM3D* also achieves higher segmentation accuracy on experimental 3D biofilm data than Cellpose[34], a state-of-the-art, CNN-based, generalist algorithm for cell segmentation, and the algorithm used by Hartmann et al.[15], a specialized algorithm designed for bacterial cell segmentation based on traditional mathematical image processing. We expect that *BCM3D*, and CNN-based single-cell segmentation approaches in general, combined with noninvasive light sheet-based fluorescence microscopy will enable accurate cell tracking over time in dense 3D biofilms. This capability will launch a new era for bacterial biofilm research, in which the emergent properties of microbial populations can be studied in terms of the fully resolved behavioral phenotypes of individual cells.

## Results
**Cell segmentation by thresholding CNN confidence maps.** CNNs have been shown to perform well on pixel-level classification tasks for both 2D and 3D data[35,36]. Bacterial biofilms, however, present a unique challenge in this context. The cell shapes to be segmented are densely packed and barely resolvable even with the highest resolution optical microscopes. In addition, living biofilms in fluorescence microscopes can only be imaged

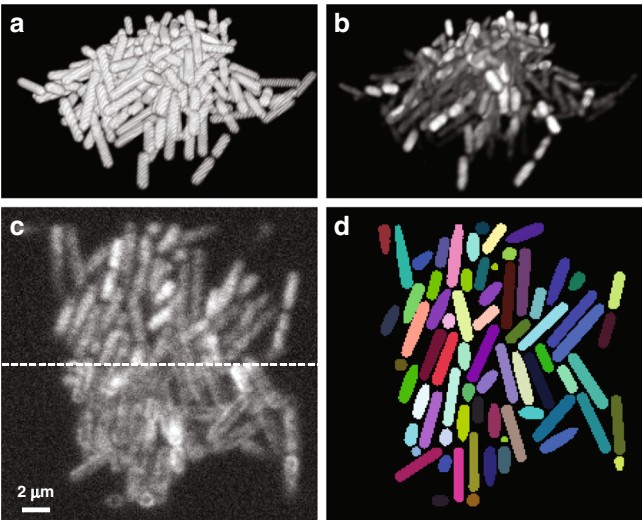

**Fig. 1 Simulation of fluorescent biofilms images and annotation maps used for CNN training. a** Representative cell arrangements obtained by CellModeller. Due to the stochastic nature of biofilm growth, different cell arrangements are obtained in each new simulation. However, cell density is reproducible for each new simulated biofilm (typically $N = 10$ different biofilm simulations are used for CNN training, see "Methods" section). **b** Simulated 3D fluorescence image based on the cell arrangements in **a**. **c** XY slice through the 3D simulated fluorescence image in **b** (upper panel shows cells expressing cytosolic fluorescent proteins, lower panel shows cells stained with membrane-intercalating dyes). **d** Ground truth cell arrangements giving rise to the image shown in **c**. Voxels are displayed as black (background), or in different colors (indicating different cells).

with low laser intensities to ameliorate phototoxicity and photobleaching concerns. Unfortunately, low-intensity fluorescence excitation also reduces the SBR in the acquired images. So far, it remains unclear to what extent single-cell segmentation approaches can accurately identify and delineate cell shapes in bacterial biofilm images obtained under low-intensity illumination conditions. To address this question, we implemented an in silico CNN training strategy and systematically evaluated its voxel-level classification (cell morphometry) and cell counting accuracies, using simulated biofilm images with cell densities and SBRs similar to those encountered in experimental data (see "Methods" section).

We compared two commonly used cell labeling approaches, namely genetic labeling through the expression of cell-internal fluorescent proteins and staining of the cell membranes using fluorescent dyes (Fig. 1). For both labeling approaches, voxel-level segmentation and cell counting accuracies, obtained by thresholding CNN confidence maps (see "Methods" section), depend mostly on cell density, whereas the SBR plays a less important role (Fig. 2a–f). For cell-internal labeling, SBRs of >1.7 and cell densities of <60% consistently produce voxel-level classification accuracies of >80% and cell counting accuracies of >95%. On the other hand, SBRs of <1.7 and cell densities of >60% lead to lower segmentation accuracies. While lower segmentation accuracies are expected for higher cell densities and lower SBRs, the sharp drop-offs observed here may indicate a fundamental performance limitation of the CNNs employed. Still, the voxel-level classification and cell counting accuracies consistently surpass previous approaches for bacterial cell segmentation for commonly encountered cell densities and SBRs. Specifically, the cell counting accuracies obtained by Hartmann et al.[15], Seg3D[37], and Yan et al.[13] quickly drop to zero as a function of increasing

Intersection-over-Union (IoU) matching threshold (a quantitative measure of cell shape similarity relative to the ground truth, see "Methods" section), indicating that cell shapes are not accurately estimated by conventional image processing approaches (Fig. 2g–i). We also evaluated the segmentation accuracy of Cellpose, a recently developed, CNN-based cellular segmentation algorithm[34]. The segmentation accuracy of Cellpose is comparable or superior to the best-performing conventional image processing approaches—a considerable achievement given that Cellpose was trained primarily on images of eukaryotic cells. However, being a pretrained generalist model, the segmentation accuracy of Cellpose is lower than the accuracy achieved by the specialist in silico-trained CNNs of BCM3D, which were trained specifically for 3D bacterial biofilm segmentation. Overall, the cell counting accuracies obtained by BCM3D are higher than other methods and remain higher even for IoU matching thresholds larger than 0.5, indicating that cell shapes are more accurately estimated by the in silico-trained CNNs.

The accuracies of single-cell shape estimation and cell counting are predominantly affected by cell density. The variation is more prominent for membrane-stained cells, because intercellular fluorescence intensity minima are less pronounced, when cell membranes are labeled and cells physically contact each other (red arrow in Fig. 2c, f). By contrast, intracellular fluorophores produce the highest intensities at the cell center, so that the gaps between cells are more readily resolvable. Also noteworthy is the sharp drop-off in segmentation accuracies for SBRs of <1.7 for all cases. In such low SBR regimes, fluorescence signals of the cells become too difficult to be distinguished from the background. As a result, the CNNs falsely identify random noisy patterns in the background as cells. In addition, thresholding of the CNN confidence maps often yields connected voxel clusters that contain multiple bacterial cells. False identification and incomplete delineation of cells cause the pronounced decrease in segmentation accuracy for SBRs of <1.7.

**Post-processing of CNN confidence maps**. To better identify individual cells in low SBR and high cell density datasets, we developed a graph-based post-processing module (see "Methods" section) that takes advantage of the fact that bacterial cell shapes are highly conserved for a given species. Briefly, we transformed the CNN cell interior confidence maps into 3D point cloud data that trace out the central axes of individual cells. This transformation was achieved by medial axis extraction using size-constrained inscribed spheres[38] (Supplementary Fig. S1a–c). Single-cell axes are then identified as linearly clustered data points by linear cuts (LCuts)—a graph-based data clustering method designed to detect linearly oriented groups of points[39]. The so-identified single-cell axes are then mapped back onto the original segmentation volumes to obtain estimates of the 3D positions, shapes, and orientations of the now separated cells (Supplementary Fig. S1d).

Post-processing with LCuts takes advantage of a priori knowledge about expected bacterial cell sizes by removing erroneously segmented volumes that are significantly smaller than the expected value and by splitting incompletely segmented volumes representing fused cells. Improvements in cell counting accuracy of up to 15% and 36% are observed for cells labeled with cytosolic fluorophores (Fig. 3a–c and Supplementary Fig. S2) and membrane-localized fluorophores (Fig. 3d–f), respectively. The more substantial improvement for membrane-stained cells is due to fact that CNNs trained on membrane-stained cells are more prone to erroneously identifying speckled background noise as fluorescence signals in low SBR images. In addition, membrane-intercalating

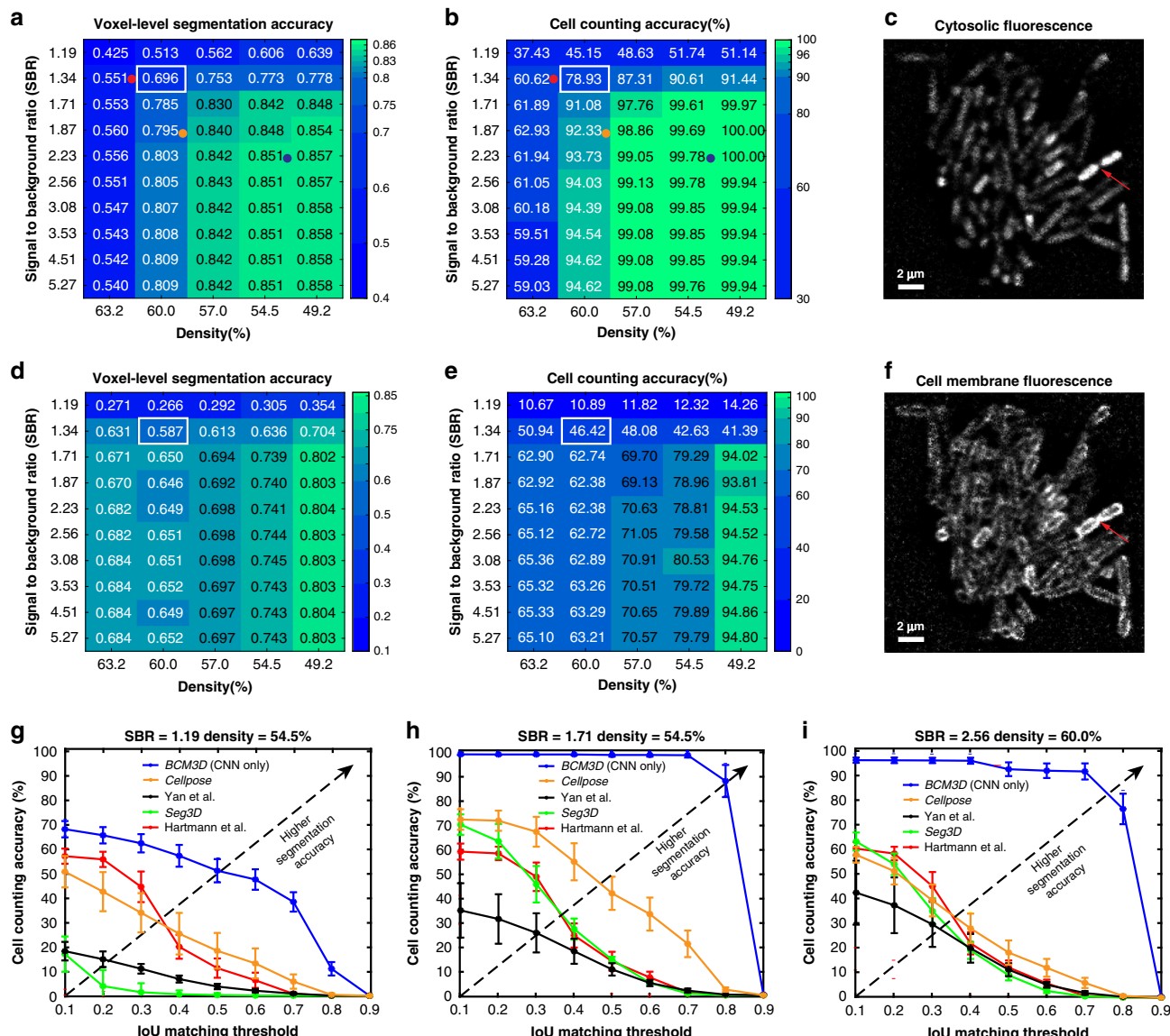

**Fig. 2 Performance of *BCM3D* using in silico-trained CNNs only on previously unseen simulated biofilm images. a** The voxel-level segmentation accuracy quantifies whether each voxel has been assigned to the correct class ("cell interior", "cell boundary", or "background"). Solid circles represent the maximum local density and average SBRs encountered in experimental datasets (red, orange, and blue: *E. coli* expressing GFP). **b** The cell counting accuracy (using an IoU matching threshold of 0.5 for each segmented object, see "Methods" section) averaged over $N = 10$ replicate datasets for cells labeled with cytosolic fluorophores. **c** Example image of cells labeled with cytosolic fluorophores (cell density = 60.0%, SBR = 1.34, indicated by white rectangle in **a** and **b**. Similar images were generated $N = 10$ times with different cell arrangements). **d** Voxel-level segmentation accuracy and **e** cell counting accuracy averaged over $N = 10$ replicate datasets for cells labeled with membrane-localized fluorophores. **f** Example image of cells labeled with membrane-localized fluorophores (cell density = 60.0%, SBR = 1.34, indicated by white rectangles in **d** and **e**. Similar images were generated $N = 10$ times with different cell arrangements). The red arrows indicate a close cell-to-cell contact point. **g–i** Comparison of segmentation accuracies achieved by conventional segmentation approaches (Hartmann et al., Seg3D, Yan et al.), Cellpose, and *BCM3D* (only using in silico-trained CNNs). Three simulated datasets (cytosolic fluorophores) with different SBRs and cell densities are shown. Segmentation accuracy is parameterized in terms of cell counting accuracy (*y*-axis) and IoU matching threshold (*x*-axis, a measure of cell shape estimation accuracy). Each data point is the average of $N = 10$ independent biofilm images. Data are presented as mean values ± one standard deviation indicated by error bars. Curves approaching the upper right-hand corner indicate higher overall segmentation accuracy, as indicated by the dashed arrows. Source data are provided as a Source data file for Fig. 2g, h, i.

fluorophores of two adjacent cells are in close proximity, making it difficult to resolve fluorescence signals from two separate cells due to spatial signal overlap (see the red arrow, Fig. 2c, f). LCuts thus provides an important benefit in improving the cell counting accuracy to an extent not achieved by currently available thresholding- or watershed-based post-processing algorithms (Supplementary Fig. S3).

**Segmentation of experimental biofilm images.** To test the performance of *BCM3D* on experimentally acquired biofilm images, we acquired time-lapse images of GFP expressing *E. coli* biofilms every 30 min for 10 h (see "Methods" section). We then manually annotated one 2D slice in the 3D images at the $t = 300$, 360, and 600-min time points (see "Methods"). When referenced to these manual segmentation results, the LCuts-processed CNN outputs

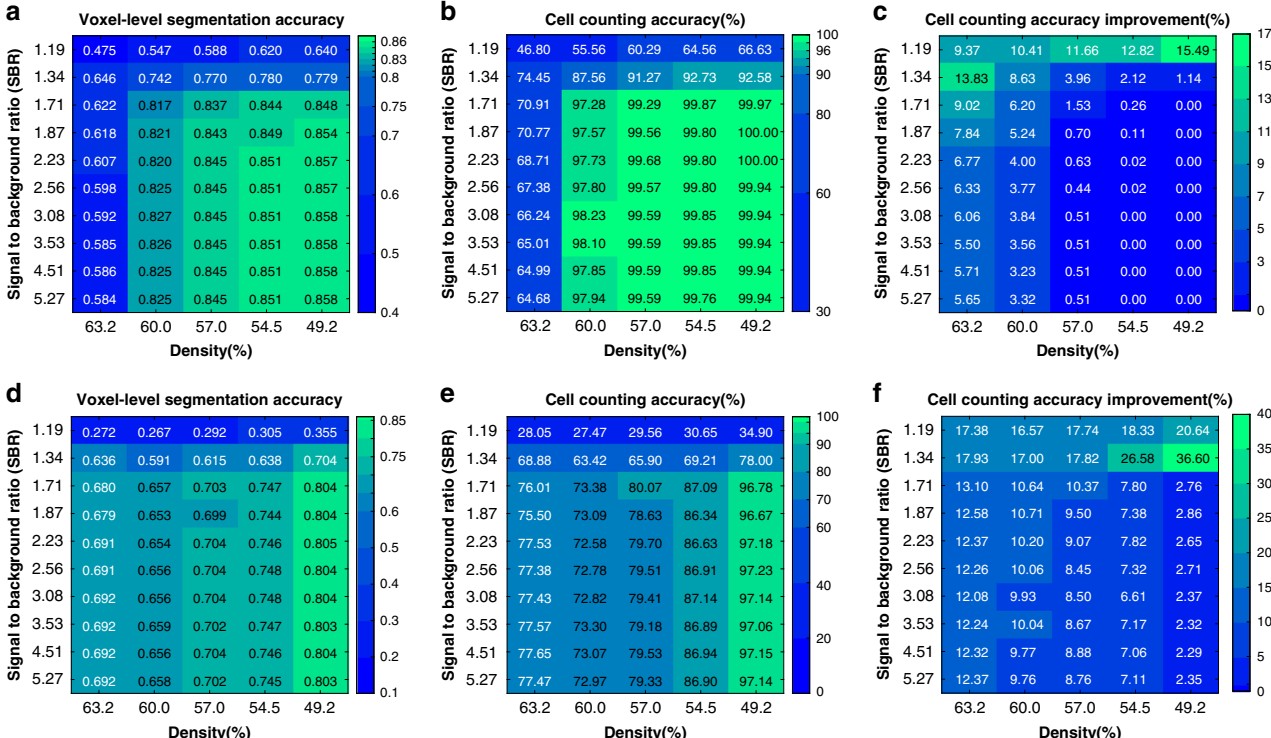

**Fig. 3 Performance of BCM3D (in silico-trained CNNs and additional post-processing by LCuts) on previously unseen simulated data. a** Voxel-level segmentation accuracy and **b** cell counting accuracy (using an IoU matching threshold of 0.5 for each segmented object) averaged over $N = 10$ replicate datasets for cells labeled with cytosolic fluorophores. **c** Improvement relative to silico-trained convolutional neural networks without post-processing. **d** Voxel-level segmentation accuracy and **e** cell counting accuracy averaged over $N = 10$ replicate datasets for cells labeled with membrane-localized fluorophores. **f** Improvements relative to silico-trained convolutional neural networks without post-processing.

consistently achieved better cell counting accuracies than conventional segmentation methods (Fig. 4 and Supplementary Fig. S4). Initially, Cellpose and the Hartmann et al. algorithm outperform the in silico-trained CNNs on two out of three of the test images ($t = 360$ and 600 min), for which our in silico-trained CNNs struggle with undersegmentation problems. However, mathematical post-processing of the CNN outputs by LCuts corrects some of these errors, so that the integrated *BCM3D* workflow achieves improved results compared to Cellpose and Hartmann et al. at each of the indicated time points. Visual inspection of the segmentation results is also informative. Cellpose accurately segments individual cells in low density regions, but suffers from oversegmentation errors in high density biofilm regions (Supplementary Fig. S4e). The Hartmann et al. algorithm provides reasonable estimates of cell positions in low and high density biofilm regions, but again struggles with cell shape estimation (Supplementary Fig. S4d and see also Fig. 2g–i). On the other hand, the integrated *BCM3D* workflow (CNN + LCuts) produces biologically reasonable cell shapes regardless of cell density (Fig. 4).

We attribute the more rapid drop-off of the cell counting accuracy as a function of increasing IoU matching threshold in Fig. 4 to the following factors. First, human annotation of experimentally acquired biofilm images differs from the ground truth segmentation masks that are available for simulated data (Supplementary Fig. S5). The shape mismatches between algorithm segmented and manually annotated cell shapes (Supplementary Figs. S5 and S6) lead to a global lowering of voxel-level segmentation accuracy, and thus a more rapid drop-off of the cell counting accuracy as a function of increasing IoU matching threshold. Because bacterial cell shapes are not

accurately captured by manual annotation (Supplementary Fig. S5), cell counting accuracies referenced to manual annotations should be compared only at low IoU matching thresholds (0.1–0.3, shaded grey in Fig. 4a–c), as also pointed out previously[40]. We also note that bacterial cells in experimental images appear motion-blurred if they are only partially immobilized, and therefore wiggle during image acquisition. Furthermore, optical aberrations and scattering effects were not included in training data simulations, which may decrease the performance of the CNNs on experimental data. Still, at IoU matching threshold <0.3, the cell counting accuracy of *BCM3D* remains above 75%, while also producing biologically reasonable cell shapes. Thus, the bacterial cell segmentation results of *BCM3D* represent a substantial improvement over other approaches (Fig. 4 and Supplementary Fig. S4).

To demonstrate that *BCM3D* can achieve similarly high segmentation accuracies for membrane-stained cells in different cellular arrangements, we analyzed a small patch of a *Myxococcus xanthus* biofilm, which was stained with the membrane-intercalating dye FM4-64 (Fig. 5a). In contrast to *E. coli* biofilms, the submerged *M. xanthus* biofilm imaged here features cells in a mesh-like arrangement with close cell-to-cell contacts, which presents a unique challenge for 3D single-cell segmentation. To obtain reference data for 3D segmentation accuracy determination, we manually annotated each *xy*, *xz*, and *yz* slice of an entire 3D image stack (Fig. 5b). When referenced to this 3D manual segmentation result, *BCM3D* (Fig. 5c) produced cell counting accuracies above 70% at low (0.1–0.3) IoU matching thresholds, whereas segmentation results obtained by conventional image processing (Hartmann et al.) and by generalist CNN-processing (Cellpose) produced cell counting accuracies <50% in the same

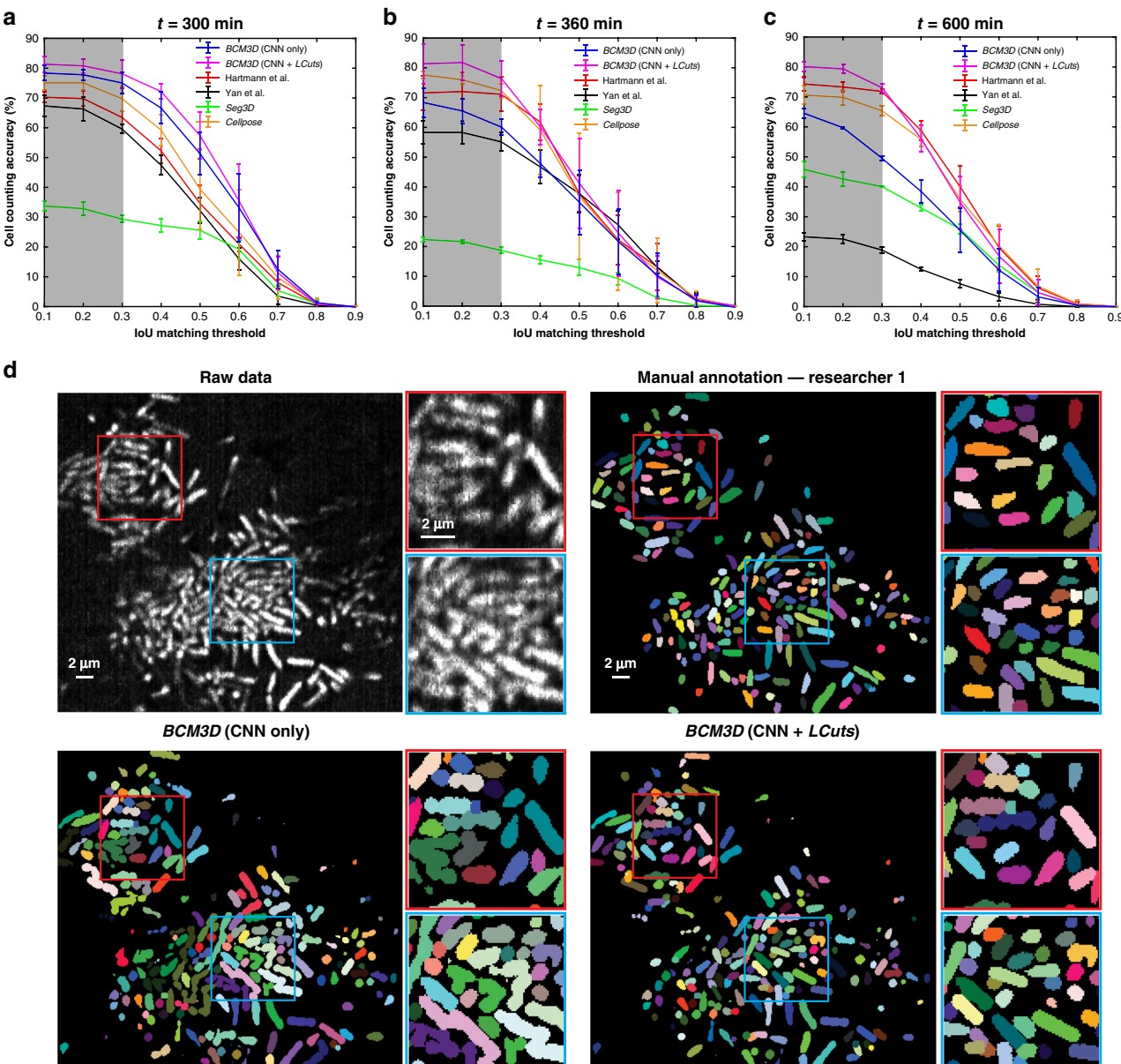

**Fig. 4 Comparison of segmentation accuracies achieved by conventional segmentation approaches (Hartmann et al., Seg3D, Yan et al.), Cellpose, and BCM3D.** The estimated SBRs are 2.2, 1.8, and 1.3, respectively. The estimated cell densities are 54.8%, 59.0%, and 64.6%, respectively. **a–c** Three experimental *E. coli* datasets (cytosolic expression of GFP) acquired at different time points after inoculation of cells. Segmentation accuracy is parameterized in terms of cell counting accuracy (y-axis) and IoU matching threshold (x-axis). Each data point is the average of the cell counting accuracies calculated using annotation maps traced by $N = 3$ different researchers. Data are presented as mean values ± one standard deviation indicated by error bars. Curves approaching the upper right-hand corner indicate higher overall segmentation accuracy. **d** Comparison of segmentation results achieved at the $t = 600$ min time point by manual annotation (shown is one of $N = 3$ researchers' annotation result, the other two annotation results are shown in Supplementary Fig. S4), and by *BCM3D* using in silico-trained CNNs only and after further refinement of CNN outputs using LCuts. Similar results were also obtained at the $t = 300$ and $t = 360$ min time points. Segmentation results of the other methods are shown in Supplementary Fig. S4. Source data are provided as a Source data file for Fig. 4a–c.

IoU matching threshold region (Fig. 5d). We note however that neither Cellpose nor the Hartmann et al. algorithms were specifically optimized/designed for segmenting membrane-stained cells. Indeed, the performance of Cellpose on this type of biofilm architecture is inferior to the results achieved using the in silico-trained CNNs of *BCM3D* alone (without using LCuts post-processing). One reason might be that the pretrained, generalist Cellpose model has not been trained sufficiently on long, thin, and highly interlaced rod-shaped cells, such as those contained in a *M. xanthus* biofilm.

**Morphological separation of mixed cell populations.** Given the improved segmentation results obtained using *BCM3D*, we reasoned that the same CNNs may have additional capacity to assign segmented objects to different cell types based on subtle morphological differences in the acquired images. Differences in the imaged cell morphologies arise due to physical differences in cell shapes (e.g., spherical vs. rod-shaped cells) or due to differences in the fluorescent labeling protocols (e.g., intracellular vs. cell membrane labeling), because fluorescence microscopes simply measure the spatial distributions of fluorophores in the sample.

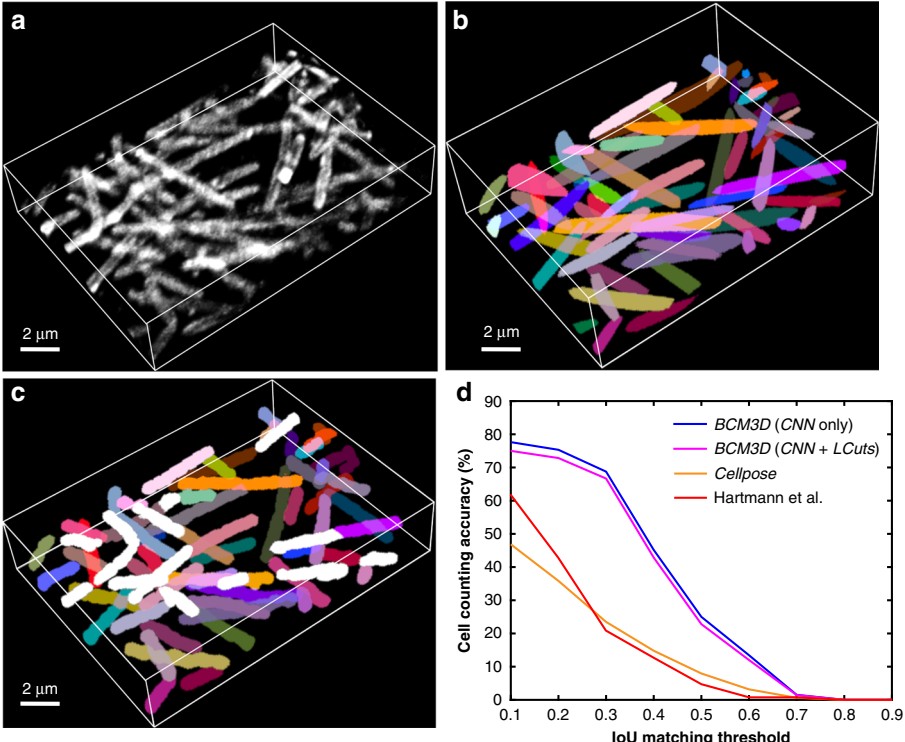

**Fig. 5 3D Segmentation accuracy evaluation using *M. xanthus* biofilm images (cell density = 36.2%, and SBR = 1.58). a** Maximum intensity projection of a 3D *M. xanthus* fluorescence image. Cells were labeled with membrane-intercalating dye, FM4-64. Similar images were obtained at $N = 120$ different time points. **b** Maximum intensity projection of the manually obtained 3D segmentation result. **c** Maximum intensity projection of a CNN-based 3D segmentation result after LCuts post-processing. Cells that can be matched with the GT are displayed in the same colors as GT or otherwise colored in white. **d** Segmentation accuracy of compared algorithms parameterized in terms of cell counting accuracy (y-axis) and IoU matching threshold (x-axis). Source data are provided as a Source data file for Fig. 5d.

The ability to separate different cell morphologies is important for the study of multispecies biofilms, where interspecies cooperation and competition dictate population-level outcomes[3,41–48]. Separation of differentially labeled cells is also important for the study of gene activation in response to cell-to-cell signaling[49]. Expression of cytosolic fluorescent proteins by transcriptional reporter strains is a widely used technique to visualize activation of a specific gene or genetic pathway in living cells. Such genetic labeling approaches can be complemented by chemical labeling approaches, e.g., using membrane-intercalating chemical dyes that help visualize cells nonspecifically or environmentally sensitive membrane dyes that provide physiological information, including membrane composition[50,51], membrane organization and integrity[52–54], and membrane potential[42,55]. Chemical and genetic labeling approaches are traditionally implemented in two different color channels. However, there are important drawbacks to using multiple colors. First and foremost, the amount of excitation light delivered is increased by the necessity to excite differently colored fluorophores, raising phototoxicity, and photobleaching concerns. Second, it takes $N$ times as along to acquire $N$-color images (unless different color channels can be acquired simultaneously), making it challenging to achieve high temporal sampling in time-lapse acquisition. For these reasons, methods that extract complementary physiological information from a single-color image are preferable.

We evaluated the ability of *BCM3D* to automatically segment and identify rod-shaped and spherical bacterial cells consistent, with shapes of *E. coli* and *Staphylococcus aureus* in simulated images (Supplementary Fig. S7). To segment cells in two-population biofilms, we trained CNNs that classify pixels into five different classes: "background", "cell interior of population 1", "cell boundary of population 1", "cell interior of population 2", and "cell boundary of population 2". Thresholding the CNNs confidence maps can achieve cell counting accuracies larger than 90% for both cell types independent of their population fractions (Fig. 6a). Post-processing of this result using LCuts improved the cell counting accuracy by <0.5% on average, indicating that under-segmented cell clusters are not prevalent in this dataset.

We next evaluated the ability of *BCM3D* to automatically segment and separate membrane-stained cells that express cytosolic fluorescent proteins from those that do not (Supplementary Fig. S8). Again, the cell counting accuracy is consistently above 80% for all tested mixing ratios (Fig. 6b). Finally, we applied *BCM3D* to experimentally acquired biofilm images of two different *E. coli* strains. Both strains were stained by the membrane-intercalating dye FM4-64, but the second strain additionally expressed GFP (Supplementary Fig. S9). The cells were homogeneously mixed prior to mounting to randomize the spatial distribution of different cell types in the biofilm (see "Methods" section). Multiple 2D slices from the 3D image stack were manually annotated and compared with the results obtained by *BCM3D*. Consistent with the single-species experimental data, a cell counting accuracy of 50% is achieved for each cell type at a 0.5 IoU matching threshold and, at lower IoU matching thresholds, the counting accuracies increased to 60–70%, (Fig. 6c, d). Thus, using appropriately trained CNNs in *BCM3D* enables automated and accurate cell type assignments based on subtle differences in cell morphologies in mixed population biofilms—a capability not available using conventional image processing methods.

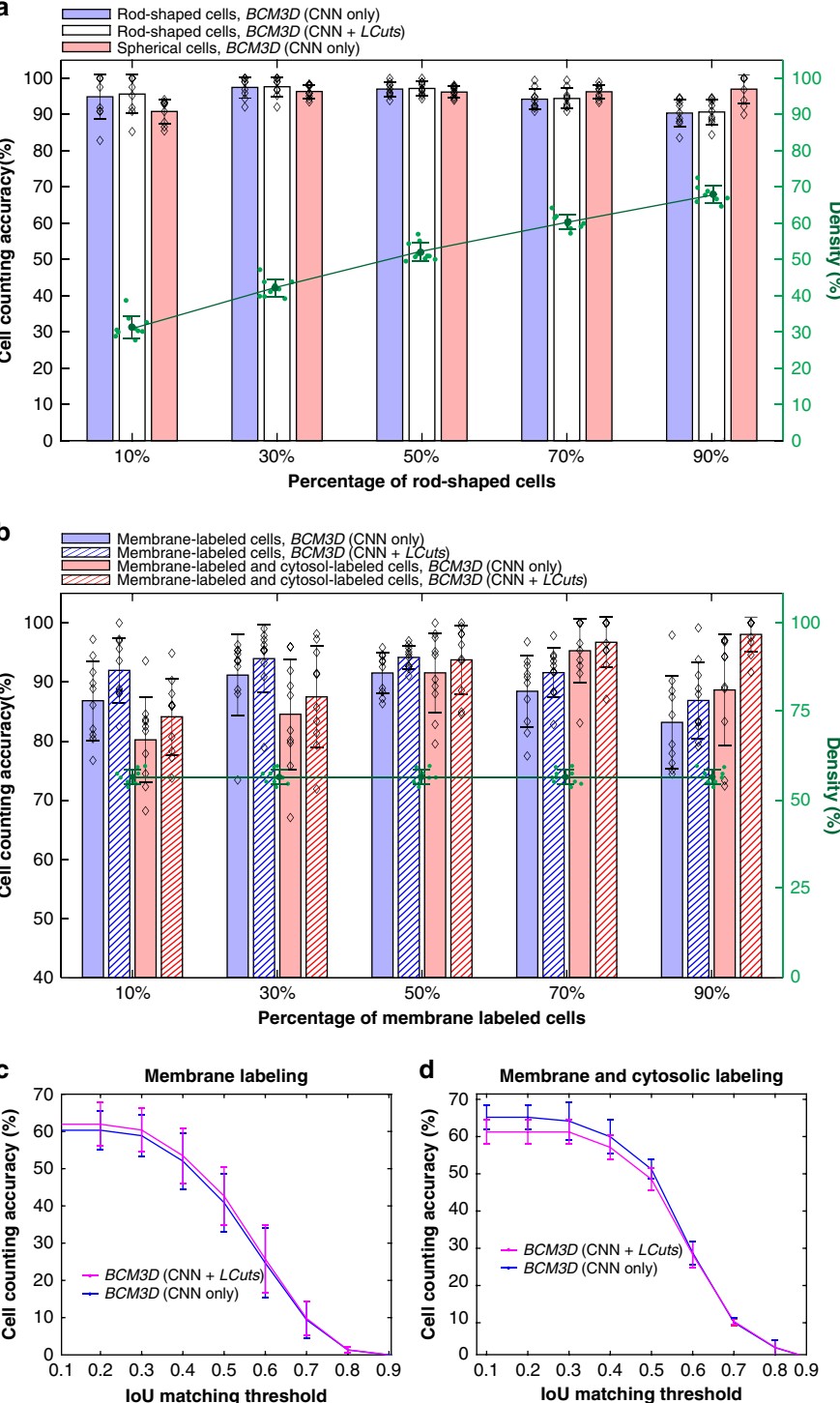

**Fig. 6 Performance of BCM3D on mixed population biofilm images. a** Cell counting accuracy of *BCM3D* on simulated images containing different ratios of rod-shaped and spherical cells. Black diamonds represent the counting accuracy for *N* = 10 independently simulated datasets. Green dots represent the cell density for each independent dataset. Error bars represent ± one standard deviation. **b** Cell counting accuracy of *BCM3D* on simulated images with different ratios of membrane-labeled, and membrane-labeled and interior fluorescent protein expressing cells. Black diamonds represent the counting accuracy for *N* = 10 independently simulated datasets. Green dots represent the cell density for *N* = 10 independent datasets. Error bars represent ± one standard deviation. **c, d** Cell counting accuracy of *BCM3D* on experimental images of **c** membrane-labeled, and **d** membrane-labeled and interior fluorescent protein expressing *E. coli* cells (mixing ratio ~1:1). Each data point is the average of the cell counting accuracies calculated using annotation maps traced by three different researchers (*N* = 3). Data are presented as mean values ± one standard deviation indicated by error bars. Source data are provided as a Source data file for Fig. 6.

## Discussion

CNNs have been successful applied to many different problems in biological image analysis, but their ability to segment individual cells in 3D and time-lapse 3D bacterial biofilm images has not yet been fully explored. Here, we demonstrated a CNN-based image analysis workflow, termed *BCM3D*, for single-cell segmentation and shape classification (morphometry) in 3D images of bacterial biofilms. In this work, we applied *BCM3D* to 3D images acquired by LLSM. However, *BCM3D* readily generalizes to 3D images acquired by confocal microscopy or advanced super-resolution microscopy modalities, provided that realistic image formation models are used to simulate the training datasets. The use of simulated training data is a major advantage of *BCM3D*, because it overcomes inconsistencies inherent in manual dataset annotation (Supplementary Figs. S5 and S6), and thus solves the problem of obtaining sufficient amounts of accurately annotated 3D image data. The ability to use simulated training data provides needed flexibility not only in terms of the microscope platform used for imaging, but also in terms of the bacterial cell shapes that are to be segmented.

We systematically investigated the advantages and limitations of *BCM3D* by evaluating both voxel- and cell-level segmentation accuracies, using simulated and experimental datasets of different cell densities and SBRs. *BCM3D* enabled accurate segmentation of individual cells in crowded environments and automatic assignments of individual cells to specific cell populations for most of the tested parameter space. Such capabilities are not readily available when using previously established segmentation methods that rely exclusively on conventional image and signal processing algorithms.

While *BCM3D* surpasses the performance of previous approaches, we stress that further improvements are possible and, for long-term, high frame-rate time-lapse imaging experiments, absolutely needed. Our systematic analysis revealed that high cell density and low SBR datasets are particularly challenging for the CNNs used in this work. Future work will therefore focus on increasing the contrast and resolution in bacterial biofilm images. While, the use of optical super-resolution modalities can provide higher spatial resolution, such resolution improvements often come at a cost of reduced image contrast and faster photobleaching/phototoxicity. Software solutions that can process images with limited resolution and low SBRs will therefore play a tremendously important role in biological imaging. *BCM3D* is a general workflow that integrates computational simulation of training data, in silico training of CNNs for a specific task or a specific cell type, and mathematical post-processing of the CNN outputs. Incorporating different training strategies and different CNNs, such as the generalist CNN used in Cellpose[34], into the *BCM3D* workflow will enable automated cross-validation of segmentation results when a ground truth or manual annotation map is not available. Furthermore, CNN-based image processing modules developed for contrast enhancement and denoising have also surpassed the performance of conventional methods based on mathematical signal processing[56–59]. Incorporating these tools into the *BCM3D* workflow promises to further improve the single-cell segmentation accuracies. We anticipate that the ability to accurately identify and delineate individual cells in dense 3D biofilms will enable accurate cell tracking over long periods of time. Detailed measurements of behavioral single-cell phenotypes in larger bacterial communities will help determine how macroscopic biofilm properties, such as its mechanical cohesion/adhesion and its biochemical metabolism, emerge from the collective actions of individual bacteria.

## Methods

**Lattice light sheet imaging of bacterial biofilms**. Fluorescence images of bacterial biofilms were acquired on a home-built LLSM. LLSM enables specimen illumination with a thin light sheet derived from 2D optical lattice[18,60].

Here, a continuous illumination light sheet was produced by a time-averaged (dithered), square lattice pattern[18], and the illumination intensity at the sample was <1 W/cm². The submicrometer thickness of the excitation light sheet is maintained over long propagation distances (~30 μm), which enables optical sectioning, and thus high resolution, high contrast imaging of 3D specimens comparable to confocal microscopy. However, fluorophore excitation by a 2D light sheet reduces phototoxicity, because each excitation photon has multiple opportunities to be absorbed by fluorophores in the excitation plane and produce in-focus fluorescence. Widefield fluorescence images corresponding to each illuminated specimen plane are recorded on a sCMOS detector (Hamamatsu ORCA Flash v2). In this work, 3D biofilm images were acquired by translating the specimen through the light sheet in 200 nm steps, using a piezo nanopositioning stage (Physik Instrumente, P-621.1CD). The data acquisition program is written in LabVIEW 2013 (National Instruments).

Ampicillin-resistant *E. coli* K12, constitutively expressing GFP[61], were cultured at 37 °C overnight in LB medium with 100 μg/ml ampicillin. Overnight cultures were diluted 100 times into the same culture medium, grown to an optical density at 600 nm (OD600) of 0.6–1.0, and then diluted by an additional factor of 10. Round glass coverslips with the diameter of 5 mm were put into a 24-well plate (Falcon) and 400 μl of cell culture was added to the well. Cells were allowed to settle to the bottom of the well and adhere to the coverslip for 1 h. The round coverslips were then mounted onto a sample holder and placed into the LLSM sample-basin filled with M9 medium. GFP fluorescence was excited using 488 nm light sheet excitation. Biofilm growth was imaged at room temperature every 30 min for a total of 20 time points. At each time point, a single 3D image stack contained 400 images, each acquired with a 15 ms exposure time to avoid motion blur.

*Myxococcus xanthus* strain LS3908 expressing tdTomato under the control of the IPTG-inducible promoter[62] and DK1622 (WT) were cultured in the nutrient-rich CYE media at 30 °C until it reached an OD600 of 0.6–1.0. Media was supplemented with 1 mM IPTG for tdTomato expressing cells. Chitosan (Thermo Fisher)-coated 5 mm round glass coverslips were prepared by incubating coverslips with 1% (w/v) chitosan (1.5% glacial acetic acid (v/v)) at room temperature for 1 h. Coverslips were then rinsed with water and placed into a 24-well plate (Falcon) with 350-400 μl of undiluted cell culture. WT cells were stained directly in the 24-well plate with 5 ng/ml FM4-64 (Thermo Fisher) dye. Cells were allowed to settle and adhere to the coverslip for 2 h. After the settling period, the coverslip was gently rinsed with CYE media to flush away unattached cells. The rinsed coverslip was then mounted onto a sample holder and placed into the LLSM sample-basin filled with MC7 starvation buffer. tdTomato and FM4-64 fluorescence was excited using 561 nm light sheet excitation. The 3D image stack contained 400 2D images. Each 2D slice was acquired with an exposure time of 30 ms.

For mixed population biofilm imaging, ampicillin-resistant *E. coli* K12, constitutively expressing GFP[61], and ampicillin-resistant *E. coli* K12, expressing mScarlet (pBAD vector, arabinose induce) were cultured separately at 37 °C overnight in LB medium with 100 μg/ml ampicillin. Overnight cultures were diluted 100 times into the same culture medium, grown to an optical density at 600 nm (OD600) of 0.6–1.0, and then diluted to an OD of 0.1. After dilution, the two strains were mixed together. Round glass coverslips with the diameter of 5 mm were put into a 24-well plate (Falcon) and 500 μl of cell culture was added to the well. Cells were allowed to settle to the bottom of the well and adhere to the coverslip for 1 h. The cell culture medium was then removed and replaced by 500 μl M9 medium containing 0.2% (w/v) arabinose. The co-culture was incubated at 30 °C overnight. Ten minutes before imaging, the co-culture was stained with 5 ng/ml FM4-64 (Thermo Fisher) dye. 3D image stacks of 20 planes with 5 ms exposure time per frame were acquired using 488 nm excitation.

**Raw data processing**. Raw 3D images were background subtracted and then deskewed and deconvolved[18,19]. The background was estimated by averaging intensity values of dark areas (devoid of cells) in the field of view. Deconvolution was performed using the Richardson–Lucy algorithm with ten iterations using experimentally measured point spread functions (PSFs) as the deconvolution kernel. The experimentally measured PSFs were obtained separately for each color channel using fluorescent beads (200 nm FluoSpheres®, Thermo Fisher) coated on a coverslip[63]. 3D images were rendered using the 3D Viewer plugin in Fiji[64] or ChimeraX[65].

**Generation of simulated biofilm images**. To generate data for training of CNNs, we computationally simulated fluorescence images of 3D biofilms, for which spatial arrangements among individual cells are known precisely and accurately. Growth and division of individual rod-shaped cells in a population were simulated using Cell-Modeller, an individual-based computational model of biofilm growth (Fig. 1a)[66]. In individual-based biofilm growth models, cells are the basic modeling units. Each cell is characterized by a set of parameters, including its 3D position, volume, and spatial orientation. All the cells in the simulated biofilm are then allowed to evolve in time according to predefined biological, chemical, and mechanical rules. For example, cells grow at a defined rate and then divide after reaching a certain volume threshold. Cellular collisions that are due to cell growth are alleviated by imposing a minimum distance criterion between cells at each time point. For our simulations, we chose cell diameter and cell length ($d$, $l$) parameters consistent with a given bacterial species, namely (1 μm, 3 μm) for *E. coli*[67], (0.7 μm, 6 μm) for *M. xanthus*[68], and (1 μm, 1 μm)

for spherically symmetric *S. aureus*[69]. While the cell volume can be readily adjusted in CellModeller, the cellular volume density, which is determined by the intercellular spacing, is not directly adjustable. We therefore adjusted the cellular volume density after each simulation by scaling the cellular positions (cell centroids), and thus the intercellular distances by a constant factor, while leaving cell sizes, shapes, and orientations unchanged. This post-processing procedure enabled simulation of the exact same 3D cell arrangements at adjustable cell volume densities.

We fluorescently labeled simulated cell volumes and surfaces according to two commonly used labeling strategies in fluorescence microscopy. To simulate expression of intracellular fluorescent proteins, the fluorescence emitters were placed at random positions within the cell volume. To simulate membrane staining, the fluorescence emitters were placed at random positions on the cell surface. Each cell contained between 500 and 1000 fluorophores to simulate expression level variations between cells, which is often observed in experimental images. Once the fluorophore spatial distributions were determined, a 3D fluorescence image (Fig. 1b) was computationally generated. Each fluorophore was treated as an isotropic point emitter, so that it would produce a diffraction-limited PSF on the detector. Experimentally measured 3D PSF shapes (see "Raw data processing" section) were used as the convolution kernel. Next, the fluorescence signal intensity was scaled by multiplying the image by a constant factor and then a constant background intensity was added to the image at ~200 photons per pixel, as measured in experimental data. This procedure enabled independent adjustments of the fluorescence signal and background to obtain SBRs consistent with experimental data. In a final step, we introduced Poisson-distributed counting noise, based on the summed background and signal intensities, as well as Gaussian-distributed camera read-out noise (experimentally calibrated for our detector at 3.04 photons per pixel on average)[70]. This resulting image data (Fig. 1c) was then processed in the same manner as experimental data (see "Raw data processing" section). In contrast to experimental data, generation of the corresponding voxel-level annotation maps is fast and error free, because the underlying ground truth cell arrangements are known a priori (Fig. 1d).

To mimic imaging of reporter gene expression in a subset of cells, we simulated biofilm images, in which all cells were stained at the cell surface (e.g., with a membrane-intercalating fluorescent dye) and a subset of cells additionally contained intracellular fluorophores (e.g., through the expression of an intracellular fluorescent protein; Supplementary Fig. S10a, b). The mixing ratios between membrane-labelled, and membrane and interior labelled cells were 10:90, 30:70, 50:50, 70:30, and 90:10. Ten different cell arrangements containing ~300 cells were simulated for each ratio. To train the CNNs (see next section), six datasets were used, all with a 50:50 mixing ratio.

To mimic imaging of cells with different morphologies, we simulated biofilms containing spherical and rod-shaped cells (Supplementary Fig. S10c, d). Cell arrangements were first simulated using rod-shaped cells and then a fraction of rod-shaped cells is replaced with spherical cells. The size of the rod-shaped cells is that of *E. coli* (~3 × 1 μm, length by diameter). The size of the spherical cells is that of *S. aureus* (~1 μm in diameter)[71]. Both cell types were labelled by intracellular fluorophores, as described above. The mixing ratios between rod-shaped and spherical cells were 10:90, 30:70, 50:50, 70:30, and 90:10. Ten different cell arrangements containing ~300 cells were simulated for each ratio. To train the CNNs (see next section), we picked one image from each mixing ratio for a total of five images.

**Training the convolutional neural networks**. We trained 3D U-Net CNNs for voxel-level classification tasks[72] within the NiftyNet platform[73] (network architecture depth 4, convolution kernel size 3, ReLU activation function, 32 initial feature maps, and random dropout of 0.5 during training). To achieve robust performance, we trained these networks using five to ten simulated biofilm images with randomly selected cell densities and SBRs (see "Generation of simulated biofilm images" section). The same raw data processing steps used for experimental data (see "Raw data processing" section) were also applied to simulated data. 3D deconvolved simulated data and their corresponding voxel-level annotations were used to train the CNNs. Each image used for training contained ~9 million voxels. We trained CNNs by classifying each voxel as "background", "cell interior", or as "cell boundary" based on the underlying cell arrangements. For mixed-species biofilms, two additional classes, "cell interior" and "cell boundary" of the second species, were used. This type of annotation scheme has been shown to increase separation of bacterial cells in 2D (ref. [74]). For data augmentation, we applied NiftyNet's built-in scaling, rotation, and elastic deformation functions. Instead of the original cross-entropy loss function combined with uniform sampling, we used the Dice loss function and "balanced sampler", so that every label has the same probability of occurrence in training. All networks were trained for 2000–3600 iterations with a learning rate of 0.0001. Using these parameters, it took ~24 h to train the CNNs on a NVIDIA Tesla V100 GPU with 16 GB memory.

**Thresholding of CNN-produced confidence maps**. Voxel-level classification by CNNs generates different confidence maps (one confidence map for each annotation class). The confidence values range between 0 and 1, and represent the confidence of assigning individual voxels to a given class. After thresholding the "cell interior" confidence map to obtain a binary image (Supplementary Fig. S11a–c), connected voxel clusters can be isolated and identified as single-cell objects using 3D connected component labeling[75]. A conservative size-exclusion

filter was applied: small objects with a volume approximately ten times less than the expected cell size were considered background noise and filtered out using an area open operator[75]. Since the cell-interior volumes do not contain the cell boundaries, we dilated each object by 1–2 voxels to increase the cell volumes using standard morphological dilation[75]. The threshold value to segment individual cell objects based on the "cell interior" confidence map was determined by plotting the overall voxel-level segmentation accuracy, quantified as the IoU value (aka Jaccard index[76]) vs. the confidence value thresholds (Supplementary Fig. S11d, e). Optimal voxel-level segmentation accuracies were consistently obtained using confidence thresholds between 0.88 and 0.94. Throughout this work, we use 0.94 for cells labeled with intracellular fluorophores and 0.88 for cells labeled with membrane-localized fluorophores.

**Post-processing of U-Net result using a refined LCuts algorithm**. Thresholding of the "cell interior" confidence map produces a binary segmentation result (background = 0, cell interior = 1), where groups of connected, nonzero voxels identify individual cells in most cases. However, when cells are touching, they are often not segmented as individuals, but remain part of the same voxel cluster (undersegmentation). On the other hand, a single cell may be erroneously split into smaller subcellular objects (oversegmentation). Finally, in datasets with low SBR, connected voxel clusters may be detected that do not correspond to cells and thus produce false positive objects (Supplementary Fig. S1a). To address these errors and improve the segmentation accuracy further, we included additional mathematical image analysis steps to post-process the CNN results and reduce under-segmentation and oversegmentation errors.

Step 1: False positive objects are identified by evaluating the coefficient of variation[77,78] for each connected voxel cluster $i$:

$$CV_i = \frac{\sigma_i}{\mu_i}, \qquad (1)$$

where $\sigma_i$ and $\mu_i$ denote the standard deviation and the mean of the intensity taken over all voxels contained in connected voxel cluster $i$. If the coefficient of variation is larger than $\rho$, then the current object will be classified as a false positive object and removed from the confidence map by setting all its voxels to zero. The removed objects will then no longer be counted when evaluating the cell counting accuracy. The value of $\rho$ is selected based on the coefficient of variation of the background. For the datasets analyzed here, this sample coefficient of variation was determined to be $\rho = 1.1$. After CV filtering, objects smaller than 25% of the expected bacterial cell size are also removed by setting its voxels to zero. The remaining connected voxel clusters are then considered for further processing (Supplementary Fig. S1a).

Step 2: To identify and delineate individual cells in the connected voxel clusters identified in the previous step, we implemented medial axis extraction using the method of inscribed spheres[38], with the constraint that the sphere radii do not exceed the expected diameter of a single bacterial cell (e.g., $d = 0.8$ μm; Supplementary Fig. S1b left). The set of $N$ inscribed spheres are tangent to the object's surface and parameterized by $(x_i, y_i, z_i; r_i < d/2)$ for $i = 1, …, N$. Determination of the $(x_i, y_i, z_i; r_i)$ coordinates is achieved using the Euclidean distance transform of the objects' boundary[79], so that the points with coordinates $(x_i, y_i, z_i)$ reliably trace out the central cell axes of individual bacterial cells (Supplementary Fig. S1b right).

Step 3: To separate different linear segments after cell axis extraction (Supplementary Fig. S1c), we used a refined version of the LCuts algorithm[39,80]. LCuts is a graph-based data clustering method designed to detect linearly oriented groups of points with certain properties. The fundamental elements of a weighted mathematical graph are nodes, edges, and edge weights. Here, the points with coordinates $(x_i, y_i, z_i)$ represent the graph nodes. Edges are the connections among nodes. Edges are assigned weights, for example, to reflect the confidence that two nodes belong to the same group. LCuts achieves grouping by assigning weights to edges in the fully connected graph to reflect the similarity between two nodes. The features of each node include its location and direction, where the location of each node is simply its Cartesian coordinates. The direction of each node is found by first determining its 5-hop neighborhood, removing nodes at large relative angles, and evaluating the major direction of the outlier removed neighborhood (Supplementary Fig. S12).

The algorithm to separate the nodes into different groups is a recursive graph cutting method[39]. Graph cuts (e.g., nCut[81]) disconnect the edges between two groups of nodes when the combined weights of these edges are minimized. The weights, between node $i$ and node $j$, are calculated as follows:

$$w_{ij} = w_D \cdot w_T, \qquad (2)$$

where

$$w_D = \begin{cases} e - D_{ij}^2/\sigma_D^2 & \text{if } D_{ij}^2 \le r \\ 0 & \text{if } D_{ij}^2 > r, \end{cases} \qquad (3)$$

$$w_T = e^{-\left(\cos(\theta_{ij}) - 1\right)^2/\sigma_T^2}. \qquad (4)$$

$w_D$ weighs the distance between two nodes and $w_T$ weighs difference between node directions. $D_{ij}$ is the Euclidean distance between node $i$ and node $j$, and $r$ is set

to eliminate edges between two far away nodes. $\theta_{ij}$ is the relative angle between the directions of nodes $i$ and $j$. $\sigma_D$ and $\sigma_T$ are adjustable parameters that control the rate of exponential decay. LCuts continues to separate groups of nodes until each group satisfies a stopping criterion. The stopping criterion is biologically inspired based on the expected length $L$ of a single bacterial cell and a group's linearity after each recursion. LCuts yields linearly oriented groups of points that trace out the central axes of individual cells (Supplementary Fig. S1c). Importantly, cell separation is achieved without having to specify the number of cells in the biofilm in advance. Furthermore, to limit the need for optimization of post-processing routines, the four adjustable parameters used in LCuts, namely cell diameter $d$, the cell length $L$, and the decay parameters $\sigma_D$ and $\sigma_T$ are chosen based on a priori knowledge about the bacterial cells under investigation. We found that the performance of *LCuts* is not sensitive to the particular values of $d$, $L$, $\sigma_D$, and $\sigma_T$ as long as they are consistent with the imaged bacterial cell sizes and shapes (Supplementary Fig. S13). Identification of single cells provided by LCuts alleviates undersegmentation errors of the CNN-based segmentation.

Step 4: The final output of linear clustering can provide length, location and orientation of each cell. Based on these linear clusters, the cellular architecture of the biofilms can be reconstructed by placing geometrical models of cells in space, as shown in Supplementary Fig. S1d. For fast computation, spherocylinders are used as the geometrical model using a radius consistent with the known sizes of bacterial cells. To further refine the cell surfaces to better align with the CNN-segmented volumes, we enclosed the inscribed spheres found in Step 2 in a convex hull (Supplementary Fig. S1d).

**Performance evaluation**. We quantified segmentation accuracy both at the cell-level (object counting) and at the voxel-level (cell shape estimation). To quantify the cell-level segmentation accuracy, we designated segmented objects as true positive (TP) if their voxel overlap with the ground truth or the manual annotation resulted in an IoU value larger than a particular IoU matching threshold. This criterion ensures one-to-one matching. A threshold of 0.5 is typically chosen when reporting single cell counting accuracy values[34,40]. We follow this convention here. If the segmented cell object could not be matched to a ground truth/manually annotated cell volume, then it was counted as a false positive (FP) and the IoU value of that segmented object was set to zero. If a ground truth/manually annotated cell volume was not identified in the image, then it was counted as false negative (FN). The cell (object) counting accuracy was then defined as TP/(TP + FP + FN). The average IoU value over all segmented objects in the image quantifies the voxel-level segmentation accuracy, i.e., the accuracy of cell shape estimation.

To evaluate the accuracy of cell segmentation on experimental data, three researchers separately traced the cell contours on experimental 2D slices by using freehand selections in Fiji ROI Manger[64]. Because human annotation is very time consuming (~50 h for a complete 3D dataset containing ~300 cells in a $22 \times 32 \times 12\ \mu m^3$ volume), one to three single 2D slices were selected for each dataset. One exception is the 3D *M. xanthus*, for which the cell outlines in all available $x$-, $y$-, and $z$-slices were traced manually (Supplementary Fig. S14a). For straight, rod-shaped cells, the centroids of the resulting 2D cell contours all fall within the cell interior volume. To group together the contours belonging to the same cells, the centroid of each contour was projected along the $x$-, $y$-, and $z$-dimension. If the projected centroid was enclosed by any other contour in a different slice, then the centroid of that contour was projected onto the plane of the initial contour. Two contours were labeled as related if they contained each other's projected centroids (Supplementary Fig. S14b). This process is repeated for all possible contour pairs and their relationship is recorded in an adjacency matrix. Next, related contours were assigned to individual cells (Supplementary Fig. S14c). To separate incorrectly grouped contours, we additionally identified clusters of centroids using the DBSCAN point clustering algorithm[82] (Supplementary Fig. S14d). In a final step, we manually removed incorrectly traced contours (Supplementary Fig. S14e). Cells are reconstructed by creating convex hulls with the grouped contours (Supplementary Fig. S14f, g). This procedure determined the approximate positions, shapes, and orientations of individual cells in the 3D biofilm.

To estimate the SBRs of both simulated and experimental images, we manually selected and determined the intensities of approximately ten "signal" and ten "background" regions in the images. We computed the SBR as the mean signal intensity divided by the mean background intensity. To estimate the local density of a biofilm, we partitioned the image into several 3D tiles of size 64 by 64 by 8 voxels. We then estimated the local density as the total cell volume contained in each tile divided by the tile volume. We calculated the mean density of the ten densest tiles to define the "local density" metric reported for each dataset in the paper. To estimate the cell density in an experimentally acquired biofilm image, the same calculations were performed on either 3D manual annotations (if available) or binary masks obtained by CNN-processing.

**Reporting summary**. Further information on research design is available in the Nature Research Reporting Summary linked to this article.

## Data availability
All data used for generating the results presented in this paper are available from the corresponding author upon request. Source data are provided with this paper.

## Code availability
The code for running all the modules of BCM3D, as well as training and test data is available at https://github.com/GahlmannLab/BCM3D.git (ref. [83]) and https://doi.org/10.5281/zenodo.4088658.

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

## Acknowledgements

This work was supported in part by the US National Institute of General Medical Sciences Grant No. 1R01GM139002 (A.G. and S.T.A.) and by a Jeffress Memorial Trust Award in Interdisciplinary Research (to A.G. and S.T.A.). M.Z. and J.W. were supported by a University of Virginia Presidential Fellowship in Data Science. Y.W. was supported in part by a Dean's M.S. Ph.D. Fellowship in Data Science. We thank Lotte Søgaard-Andersen, Larry Shimkets, Huiwang Ai, and Ingmar Riedel-Kruse for providing bacterial strains used in this work. We thank Karsten Siller, computational research consultant at the University of Virginia, for help with utilizing the Universities super-computing resources. We thank Knut Dresher and Eric Jelli for providing the segmentation results using the Hartmann et al. algorithm.

## Author contributions

M.Z., J.Z., Y.W., J.W., S.T.A., and A.G. designed research; M.Z., J.Z., Y.W., J.W., and A.M.A. performed research; M.Z., J.Z., Y.W., J.W., S.T.A., and A.G. analyzed data; and M.Z., J.Z., Y.W., J.W., S.T.A., and A.G. wrote the paper.

## Competing interests

The authors declare no competing interests.
