## [Peer Review File · Nature Communications]

Reviewers' Comments:

Reviewer #1:

Remarks to the Author:

This paper describes a computational method for achieving good segmentation performance on bacterial biofilms composed of cells of various shapes. The approach is well justified, well executed and highly successful. It follows a modern approach of 1) simulating lots of training data + 2) training a deep CNN + 3) post-processing the results of the CNN. The simulation can be adjusted by the user to match the parameters of the real data the user is interested in segmenting, which gives this method more flexibility than a lot of other CNN-based approaches. The approach is shown to outperform older, non-CNN based methods by a wide margin. The authors even include comparisons to the very recently released Cellpose package, showing that their method consistently works as well as or better for the specific types of images used in this paper.

Generally speaking, I like everything about this paper. The thing I like least, that I would like to see some changes to in a revision, relates to the postprocessing workflow that follows the pixel predictions from the deep CNNs. This is always a point of concern for CNN-based methods, because the CNN is a relatively elegant, end-to-end trained method, while the post-processing tends to be idiosyncratic and manually optimized to the specific data we are seeing. It is difficult to know if this approach will work on other types of data, mainly because of this post-processing step. I think it's important to clearly state if all the hyper-parameters in the post-processing were chosen on validation data, as opposed to on the test data on which results are reported. If they were optimized on the test data, then the optimization needs to be re-done on a separate validation dataset and the best set of hyper-parameters be picked automatically with grid searches, not by a human with subjective biases.

The next part of my concerns relates to the reproducibility of this post-processing step. It doesn't help that it's in Matlab, which is closed source and unavailable to a lot of people. The rest of the code used is open source, so I recommend the authors consider rewriting their post-processing step in Python. Even better, why not use an existing package that can do the post-processing for you? The first things that come to mind are ilastik and CellProfiler, but there are many others too. A watershed based algorithm seems like it could deal pretty well with the merge problems generated by the soft pixel predictions of the CNN. Perhaps I am wrong, and the L-Cuts algorithm is really needed, but in that case I'd still like to see it shown in the paper that a simpler post-processing method was insufficient.

Marius Pachitariu

Reviewer #2:

Remarks to the Author:

The article by Zhang et al. presents a new segmentation workflow consisting of a convolutional neural network whose results are post-processed by a graph-based data clustering method developed by the same group. The authors show that they can segment bacteria in biofilms with the segmentation accuracy depending on density and signal to background ratio (SBR). What makes the article interesting is not only the performance but also that they used simulations to train the CNN and thus can circumvent the laborious collection of training sets.

Detailed comments

The authors mention depth dependent aberrations only in so far as stating that they don't consider them. But a comparison of simulations and experiments could show how strong that influence is. Does segmentation accuracy change with biofilm depth? Presumably yes if the SBR changes.

375/6: An IoU of 0.5 seems low? It would be instructive if the authors could produce a distribution of IoU values to justify that cutoff and indicate how strongly that cutoff influences the correct segmentation of cells. The authors already show graphs in Figs S2 and S5 to address this issue. But it seems to reach a high confidence level one would need an IoU significantly higher than 0.5? And why does the IoU value drop at high confidence thresholds? In other words a confidence threshold of 0.65 and 0.99 have the same IoU. So the IoU value cannot capture the quality of segmentation unambiguously?

The fact that voxel IoU and cell number accuracy is not the same, points to problems with shape determination. Perhaps authors can discuss this problem in more detail and potentially use erosion/dilation algorithms to better adapt shapes? At the moment the authors discuss this issue only in relation to other algorithms.

Could the authors provide difference maps between a) simulations and segmentation, b) between manual annotation of simulations and automatic segmentation? This would show on one side the quality of the segmentation but also show how manual segmentation is comparing to automatic segmentation when done on the same ground truth (see lines 550-561 where the authors already hint at that problem). Also in Fig. S7 such a difference map would be very helpful.

Fig. 5: there are strong differences between manual and BCM3D segmentation. Again a comparison on simulations might be good as a ground truth for comparison would be available. But it is also clear that the manual segmentation leads in general to smaller cells. Difference maps to the raw data would be very interesting.

528: is it clear when Cellpose or BCM3D is better? On average better is not a strong statement.

617: independent of population fraction? But that must be density dependent? Please clarify.

Reviewer #3:

Remarks to the Author:

In this manuscript the authors introduce a biofilm image analysis tool that combines machine learning and improved image analysis to resolve individual cells within biofilms.

There are definite strengths and weakness. The strengths include a fairly well written description of the methodology and the careful comparison of the authors program vs other established image analysis programs/approaches that are being currently used in the field. Weaknesses include limited impact- the manuscript describes an image analysis program that is better than other existing programs, but its not clear to me how significant this improvement means to the field. Additionally, the manuscript is a methods only paper, not addressing any biological question or demonstrating how the authors software could shed light on this question where other programs fall short.

Response to Reviewer Comments

Note to all reviewers:

After we posted our manuscript on bioRxiv, we received some feedback and suggestions from Prof. Knut Drescher, the corresponding author of *BiofilmQ*¹. He pointed out that *BiofilmQ* was not specifically designed for single-cell segmentation and suggested that we instead compare our results to those obtained by an algorithm developed by Hartmann *et al.*². The Hartmann *et al.* algorithm is a highly-refined bacterial cell segmentation algorithm based on traditional mathematical image processing routines. Because the source code of the Hartmann *et al.* algorithm is not yet publicly available, Prof. Drescher offered to run the algorithm on simulated and experimental images that we would provide to his lab. We found that the Hartmann *et al.* algorithm indeed outperformed *BiofilmQ* and other algorithms based on traditional mathematical image processing and it can therefore be regarded as the state-of-the-art algorithm in this category. We have therefore replaced any references pertaining to and any results obtained with *BiofilmQ* with those of the Hartmann *et al.* algorithm. The Drescher lab is now acknowledged for their contribution in the acknowledgement section of the revised manuscript. This change does not affect the conclusions we draw, but instead strengthens the significance of *BCM3D* as a more accurate and versatile segmentation approach.

Reviewer #1:

This paper describes a computational method for achieving good segmentation performance on bacterial biofilms composed of cells of various shapes. The approach is well justified, well executed and highly successful. It follows a modern approach of 1) simulating lots of training data + 2) training a deep CNN + 3) post-processing the results of the CNN. The simulation can be adjusted by the user to match the parameters of the real data the user is interested in segmenting, which gives this method more flexibility than a lot of other CNN-based approaches. The approach is shown to outperform older, non-CNN based methods by a wide margin. The authors even include comparisons to the very recently released Cellpose package, showing that their method consistently works as well as or better for the specific types of images used in this paper. Generally speaking, I like everything about this paper.

Reply: We thank the reviewer for this positive overall assessment.

Comment 1.1

The thing I like least, that I would like to see some changes to in a revision, relates to the postprocessing workflow that follows the pixel predictions from the deep CNNs. This is always a point of concern for CNN-based methods, because the CNN is a relatively elegant, end-to-end trained method, while the post-processing tends to be idiosyncratic and manually optimized to the specific data we are seeing. It is difficult to know if this approach will work on other types of data, mainly because of this post-processing step. I think it's important to clearly state if all the hyper-parameters in the post-processing were chosen on validation data, as opposed to on the test data on which results are reported. If they were optimized on the test data, then the optimization needs

to be re-done on a separate validation dataset and the best set of hyper-parameters be picked automatically with grid searches, not by a human with subjective biases.

Reply:

To limit the need for manual (idiosyncratic) optimization of postprocessing routines, the three adjustable parameters used in *LCuts* are NOT determined through data-driven optimization. Instead, these parameters are chosen based on a priori knowledge regarding the bacterial cells under investigation. These parameters are the cell diameter d , the cell length L , and the distance decay parameter σ_D . The cell diameters and lengths can be readily estimated based on images of dispersed bacterial cells and the distributions of these parameters over a given cell population is therefore well-defined for many bacterial species. If they are not available from literature sources, then the distributions can be readily obtained through a separate experiment in which biofilm cells are dispersed and then imaged in isolation. Such experiments are routine in microbiology. Here, we use the median value of the cell diameter to select d . This constrains the radii of the inscribed spheres to the interval $[d/2 - 0.2, d/2 + 0.2]$ μm for medial axis extraction (postprocessing step 2). Similarly, for linear clustering (postprocessing step 3), we constrain the cell lengths to be less than the maximum expected cell length L in the population. The distance decay parameter σ_D was chosen as the maximum cell radius.

We have included an additional section in the manuscript starting on Lines 364:

*“Furthermore, to limit the need for optimization of postprocessing routines, the four adjustable parameters used in *LCuts*, namely cell diameter d , the cell length L , and the decay parameters σ_D and σ_T are chosen based on a priori knowledge about the bacterial cells under investigation. We found that the performance of *LCuts* is not sensitive to the particular values of d , L , σ_D and σ_T as long as they are consistent with the imaged bacterial cell sizes and shapes (**Figure S4**). Identification of single cells provided by *LCuts* alleviates under-segmentation errors of the CNN-based segmentation.”*

Figure S4 validates our parameter selection by showing the results of a grid search on 20 randomly chosen, simulated datasets of low SBR and/or high cell density, for which post-processing is required to segment single cells. The simulated cells have a median diameter of 0.8 μm and vary in length from 2 to 6 μm .

Figure S4. Validation of parameter selection for *LCuts* postprocessing by grid search. Shown is the cell counting accuracy averaged over 20 randomly chosen, simulated datasets of low SBR and/or high cell density, for which post-processing is required. **(a)** Average cell counting accuracy as a function of cell diameter $d \in [0.4, 1.2] \mu\text{m}$ and cell length $L \in [2, 9] \mu\text{m}$ at a fixed $\sigma_D = 0.5 \mu\text{m}$ and $\sigma_T = 0.2$. **(b)** Average cell counting accuracy as a function of $\sigma_D \in [0.1, 0.8] \mu\text{m}$, and $\sigma_T \in [0.05, 0.6]$ with fixed $(d, L) = (0.8, 4.5) \mu\text{m}$. The cell counting accuracy is largely unaffected by variations in d , L , σ_D and σ_T and robustly remains above 70% for biologically reasonable parameter values, such as $d \sim 0.8 \mu\text{m}$, cell length $L \sim 6 \mu\text{m}$, for *E.coli*-like cell shapes. We also choose $\sigma_D = d/2$ and $\sigma_T = 0.2$, so that edges between nodes separated by more than a cells radius or with relative angles $>30^\circ$ are weighted down.

Comment 1.2

The next part of my concerns relates to the reproducibility of this post-processing step. It doesn't help that it's in Matlab, which is closed source and unavailable to a lot of people. The rest of the code used is open source, so I recommend the authors consider rewriting their post-processing step in Python. Even better, why not use an existing package that can do the post-processing for you? The first things that come to mind are *ilastik* and *CellProfiler*, but there are many others too. A watershed based algorithm seems like it could deal pretty well with the merge problems generated by the soft pixel predictions of the CNN. Perhaps I am wrong, and the *L-Cuts* algorithm is really needed, but in that case I'd still like to see it shown in the paper that a simpler post-processing method was insufficient.

Reply:

We agree with the reviewer that newly developed algorithms should be available in an open source format. We would like to point out that, while MATLAB itself is a commercial closed source platform, our MATLAB code is open source and executable on Octave, which is an open-source platform. We are however committed to providing a Python version of our code in our Github repository in the future to ensure broader adaption in the field.

We now include a new **Figure S7**) that compares *LCuts* to other open source post-processing algorithms, namely the hysteresis thresholding-based algorithm of *Ilstik* and

the watershed-based pipeline of *CellProfiler*. For the test data, we again chose the same 20 simulated datasets used in **Figure S4**. The results show that the hysteresis thresholding-based algorithm improves the cell counting accuracy by less than 6% on average. On the other hand, the watershed-based pipeline used by *CellProfiler* decreases the cell counting accuracy in many cases, which is primarily due to oversegmentation. Among the three methods tested, *LCuts* provides the highest improvement in cell counting accuracy (>12% on average for IoU matching thresholds less than 0.6). These data indicate that simpler post-processing methods commonly used by researchers do not provide the performance boost of *LCuts*. Therefore, the application of *LCuts* is indeed necessary. **Figure S7** is referenced on Line 518 of the manuscript:

“LCuts thus provides an important benefit in improving the cell counting accuracy to an extent not achieved by currently available thresholding- or watershed-based post-processing algorithms (Figure S7).”

Figure S7. Comparison of *LCuts* to commonly used image post-processing methods. Shown is the cell counting accuracy averaged over 20 randomly chosen, simulated datasets of low SBR and/or high cell density, for which post-processing is required. The hysteresis thresholding-based algorithm of *Ilastik*³ improves the cell counting accuracy by less than 6% on average for IoU matching thresholds less than 0.6. On the other hand, the watershed-based pipeline used by *CellProfiler* 3.0⁴ provides negligible improvements and even decreases the average cell counting accuracy in many cases. This decrease is primarily due to oversegmentation. Among the three methods tested, *LCuts* provides the highest improvement in cell counting accuracy (>12% on average for IoU matching thresholds less than 0.6).

Reviewer #2:

The article by Zhang et al. presents a new segmentation workflow consisting of a convolutional neural network whose results are post-processed by a graph-based data clustering method developed by the same group. The authors show that they can segment bacteria in biofilms with the segmentation accuracy depending on density and signal to background ratio (SBR). What makes the article interesting is not only the performance but also that they used simulations to train the CNN and thus can circumvent the laborious collection of training sets.

Reply: We thank the reviewer for this positive overall assessment.

Comment 2.1

The authors mention depth dependent aberrations only in so far as stating that they don't consider them. But a comparison of simulations and experiments could show how strong that influence is. Does segmentation accuracy change with biofilm depth? Presumably yes if the SBR changes.

Reply: Yes, the segmentation accuracy does change with biofilm depth, because the SBR decreases. The Figure below shows a decrease in SBR and an associated decrease in cell segmentation accuracy as a function of imaging depth. The images are of a thick 3D biofilm of spherical *Myxococcus xanthus* spores. *M. xanthus* spores are highly scattering due to their polysaccharide spore coats, so the dependence of SBR on imaging depth is very pronounced. We chose not to include this Figure in the Supporting Information, because a thorough characterization of segmentation performance as a function of imaging depth is species-specific. Instead, we chose to characterize segmentation performance as a function of SBR which is a more general metric.

Comment 2.2

375/6: An IoU of 0.5 seems low? It would be instructive if the authors could produce a distribution of IoU values to justify that cutoff and indicate how strongly that cutoff influences the correct segmentation of cells.

Reply: The choice of IoU matching threshold is arbitrary. A common practice in the field is to choose an IoU matching threshold of 0.5 when comparing single values of cell counting accuracy^{5,6}. We follow this convention in our manuscript as well. The more important point however is that cell counting accuracy is a function of IoU matching threshold and therefore this functional dependence is shown in many of our Figures, namely **Figure 3ghi**, **Figure 5abc**, **Figure 6d**, **Figure 7cd**, and **Figure S6**.

We have edited the text to clarify this point (Lines 380):

“We quantified segmentation accuracy both at the cell-level (object counting) and at the voxel-level (cell shape estimation). To quantify the cell-level segmentation accuracy, we designated segmented objects as true positive (TP) if their voxel overlap with the ground truth or the manual annotation resulted in an IoU value larger than a particular IoU matching threshold. This criterion ensures one-to-one matching. A threshold of 0.5 is typically chosen when reporting single cell counting accuracy values^{5,6}. We follow this convention here.”

Comment 2.3

The authors already show graphs in Figs S2 and S5 to address this issue. But it seems to reach a high confidence level one would need an IoU significantly higher than 0.5? And why does the IoU value drop at high confidence thresholds? In other words, a confidence threshold of 0.65 and 0.99 have the same IoU. So the IoU value cannot capture the quality of segmentation unambiguously?

Reply: We thank the reviewer for highlighting this possible point of confusion. Indeed, there are two different thresholds used in our paper.

The first threshold, the confidence value threshold, is used to assign voxels from the CNN output (containing confidence values ranging from 0 to 1) to either ‘cell interior’ or ‘background’ classes. This step will result in a binary image, which is then used for further processing, as explained in the manuscript (Lines 270ff and 287ff). The selection of the confidence value threshold is based on the data shown in **Figure S2**, which shows the voxel-level segmentation accuracy of the entire image as a function of the confidence value threshold. The reason why a confidence value threshold of 0.65 and 0.99 give a similar voxel-level segmentation accuracy is as follows: Too low confidence thresholds, such as 0.65, produce many connected objects as well as many artificial objects. This reduces the voxel-level segmentation accuracy. Too high confidence value thresholds, such as 0.99, filter out too many (true positive) cells. This also reduces the voxel-level segmentation accuracy. The optimal voxel-level segmentation accuracy is consistently obtained using confidence value thresholds between 0.88 and 0.94. Throughout this work, we use 0.88 for cells labeled with intracellular fluorophores and 0.94 for cells labeled with membrane-localized fluorophores.

We have edited the sentences on Line 279ff to clarify this point:

“The threshold value to segment individual cell objects based on the ‘cell interior’ confidence map was determined by plotting the overall voxel-level segmentation accuracy, quantified as the Intersection-over-Union value (IoU value, aka Jaccard index²) versus the confidence value thresholds (Figure S2). Optimal voxel-level segmentation accuracies were consistently obtained using confidence thresholds between 0.88 and 0.94. Throughout this work, we use 0.94 for cells labeled with intracellular fluorophores and 0.88 for cells labeled with membrane-localized fluorophores.”

We also changed the x-axis label of **Figure S2** to “Confidence value thresholds” and the y-axis label to “Voxel-level segmentation accuracy” to avoid confusion with the second threshold used in our paper, the IoU matching threshold. The IoU matching threshold is used to quantify the cell counting accuracy, as explained in section **Performance Evaluation** (Lines 379ff).

Comment 2.4

The fact that voxel IoU and cell number accuracy is not the same, points to problems with shape determination. Perhaps authors can discuss this problem in more detail and potentially use erosion/dilation algorithms to better adapt shapes? At the moment the authors discuss this issue only in relation to other algorithms.

Reply: The reviewer is correct that voxel-level segmentation accuracy and cell counting accuracy are not the same. As explained in **Performance Evaluation** (Lines 379ff), these two metrics quantify segmentation accuracy respectively at the voxel-level (cell shape estimation) and at the cell-level (object counting). We disagree, however, that the results point to problems with shape determination. Cell shapes are estimated accurately if the voxel-level segmentation accuracy is high. Single cells are accurately segmented as individual objects, if the cell counting accuracy is high. Both of these metrics need to be high to accurately segment single cell objects AND their shapes, as indicated by the dashed arrows in Figures 3ghi.

We also note that dilation is used in our workflow after thresholding the CNN confidence maps and object identification to increase cell size, as explained on Lines 277ff.

Comment 2.5

Could the authors provide difference maps between a) simulation GT and segmentation, b) between manual annotation of simulations and automatic segmentation? This would show on one side the quality of the segmentation but also show how manual segmentation is comparing to automatic segmentation when done on the same ground truth (see lines 550-561 where the authors already hint at that problem). Also in Fig. S7 such a difference map would be very helpful.

Reply: We thank the reviewer for the suggestion to visually assess the quality of our segmentation and manual annotation. Indeed, the segmentation accuracy metrics are lower, when they are evaluated with respect to the manual annotation result instead of the ground

truth. As the reviewer noticed, we already hinted at this problem in the manuscript (now Lines 565-573) To more thoroughly quantify this effect, the revised manuscript now includes a new **Figure S9** (shown below). The data in **Figure S9** substantiates our claim that manual annotations don't determine cell shapes as reliably as the ground truth. (Please also see our Reply to **Comment 2.6** below).

In addition to **Figure S9**, we have also made a slice-by-slice video showing differences in segmentation and ground truth for the mixed-shape biofilm shown in **Figure S11**.

Figure S9 (a) Fluorescence image slice of a 3D simulated biofilm and (b) the corresponding ground truth (GT). (c) The fluorescence image slice shown in (a) masked by its corresponding GT shown in (b). The fluorescence is not completely masked because of the diffraction-limited resolution of light microscopy. (d) Fluorescence image slice of the same simulated biofilm masked by the *BCM3D* segmentation result. (e) Absolute value of the difference image between the GT and the *BCM3D* segmentation result. White pixels indicate regions where the two masks do not agree. (f) Absolute value of the difference

image between a manual annotated mask (from researcher 3) and the *BCM3D* segmentation result. **(g)** Fluorescence image slice of the same simulated biofilm masked by the manual annotation result. Researcher 3 chose to draw larger cell boundaries to mask more of the fluorescence intensity. **(h)** Absolute value of the difference image between the GT and the manually annotated mask. White pixels indicate regions where the two masks do not agree. **(I)** Segmentation accuracy achieved by manual annotation performed by three different researchers. Segmentation accuracy is parameterized in terms of cell counting accuracy (y axis) and IoU matching threshold (x axis, a measure of cell shape estimation accuracy). Curves approaching the upper right-hand corner indicate higher overall segmentation accuracy with respect to the ground truth. While IoU matching thresholds <0.3 yield good cell counting accuracies, the cell counting accuracy sharply decreases for IoU matching thresholds >0.3 , because manually annotated cell shapes differ from the ground truth cell shapes.

Comment 2.6

Fig. 5: there are strong differences between manual and *BCM3D* segmentation. Again a comparison on simulations might be good as a ground truth for comparison would be available. But it is also clear that the manual segmentation leads in general to smaller cells. Difference maps to the raw data would be very interesting.

Reply: The reviewer is correct that there are differences between manual annotation and *BCM3D* segmentation. These differences are now highlighted in the new **Figure S9**, (see response to previous comment). (Unfortunately), manual annotation masks are the only references to which segmentation results of experimental images can be compared. The manual annotation and *BCM3D* segmentation mismatch is also reflected in the fact that the curves in **Figure 5abc** and **Figure S9i** do not approach the upper right hand corner. We have edited the manuscript and added a new **Figure S10** to better highlight and clarify this issue (Lines 565ff):

*“We attribute the more rapid drop-off of the cell counting accuracy as a function of increasing IoU matching threshold in **Figure 5** to the following factors. First, human annotation of experimentally acquired biofilm images differs from the ground truth segmentation masks that are available for simulated data (**Figure S9**). The shape mismatches between algorithm segmented and manually annotated cell shapes (**Figure S9** and **S10**) lead to a global lowering of voxel-level segmentation accuracy and thus a more rapid drop-off of the cell counting accuracy as a function of increasing IoU matching threshold. Because bacterial cell shapes are not accurately captured by manual annotation (**Figures S9**), cell counting accuracies referenced to manual annotations should be compared only at low IoU matching thresholds (0.1-0.3), as also noted previously⁵.”*

Figure S10 (a) Fluorescence image slice of the 3D *E. coli* biofilm shown in **Figure 5c** masked by the *BCM3D* segmentation result. (b) Fluorescence image slice of the 3D *E. coli* biofilm shown in **Figure 5c** masked by manual annotation. (c) Absolute value of the difference image between manual annotation and *BCM3D* segmentation mask. Black pixels indicate image regions where the two masks agree and white pixels indicate image regions where the two masks differ.

Comment 2.7

528: is it clear when *Cellpose* or *BCM3D* is better? On average better is not a strong statement.

Reply: We thank the reviewer for pointing out the ambiguity related to the term “on average”. The numerically quantified performance of *BCM3D* is slightly better than other state-of-the-art methods for this particular *E. coli* biofilm. However, the qualitative performance is substantially better than other state-of-the-art methods. We have edited the manuscript text to clarify this point (Lines 541ff):

“However, mathematical post-processing of the CNN outputs by LCuts corrects some of these errors, so that the integrated BCM3D workflow achieves improved results compared to Cellpose and Hartmann et al. at each of the indicated time points. Visual inspection of the segmentation results is also informative. Cellpose accurately segments individual cells in low density regions, but suffers from oversegmentation errors in high density biofilm regions (Figure S8e). The Hartmann et al. algorithm provides reasonable estimates of cell positions in low and high density biofilm regions, but again struggles with cell shape estimation (Figure S8d, see also Figure 3g-i). On the other hand, the integrated BCM3D workflow (CNN + LCuts) produces biologically reasonable cell shapes regardless of cell density (Figure 5).”

We also note that, the numerically quantified performance of *BCM3D* is substantially better than other state-of-the-art methods for the simulated dataset in **Figure 3** and **4** as well as the membrane-labeled *M. xanthus* cells in **Figure 6**. We have edited the manuscript text to clarify this point as well (Lines 581ff):

“To demonstrate that BCM3D can achieve similarly high segmentation accuracies for membrane-stained cells in different cellular arrangements, we analyzed a small patch of a M. xanthus biofilm, which was stained with the membrane intercalating dye FM4-64 (Figure 6a). In contrast to E. coli biofilms, the submerged M. xanthus biofilm imaged here features cells in a mesh-like arrangement with close cell-to-

cell contacts, which presents a unique challenge for 3D single-cell segmentation. To obtain reference data for 3D segmentation accuracy determination, we manually annotated each xy, xz, and yz slice of an entire 3D image stack (Figure 6b). When referenced to this 3D manual segmentation result, BCM3D (Figure 6c) produced cell counting accuracies above 70% at low (0.1-0.3) IoU matching thresholds, whereas segmentation results obtained by conventional image processing (Hartmann et al.) and by generalist CNN-processing (Cellpose) produced cell counting accuracies <50% in the same IoU matching threshold region (Figure 6d). We note however that neither Cellpose nor the Hartmann et al. algorithm were specifically optimized/ designed for segmenting membrane-stained cells. Indeed, the performance of Cellpose on this type of biofilm architecture is inferior to the results achieved using the in silico-trained CNNs of BCM3D alone (without using LCuts post-processing). One reason might be that the pre-trained, generalist Cellpose model has not been trained sufficiently on long, thin, and highly interlaced rod-shaped cells, such as those contained in a M. xanthus biofilm.”

Comment 2.8

617: independent of population fraction? But that must be density dependent? Please clarify.

Reply: The reviewer is correct that segmentation accuracy is cell density-dependent. That fact is made clear by the data shown in **Figure 3** and **Figure 4**. For the simulation of the two-population biofilm images, we chose a SBR (2.56), for which the segmentation accuracy varies little with cell density (unless cell density is extremely high >60%).

The cell densities vary slightly for the biofilm data used in **Figure 7a**, because cell position are first simulated using rod shaped cells. A predetermined fraction of rod-shaped cells is then replaced with spherical cells. These replacements generate unoccupied volumes in the biofilm. The reason for this ad-hoc approach is that *CellModeller* and similar freely available individual-based models do not allow for simulations of mixed cell shapes. We have corrected the manuscript text to clarify this point (Lines 241 ff):

“Cell arrangements were first simulated using rod shaped cells and then a fraction of rod-shaped cells is replaced with spherical cells.”

We have now added a secondary axis label to **Figure 7ab** to show the average cell densities at each population mixing ratio. We also specifically state the simulated cell density of the biofilm shown in **Figure S11**.

Reviewer #3:

In this manuscript the authors introduce a biofilm image analysis tool that combines machine learning and improved image analysis to resolve individual cells within biofilms.

Comment 3.1

There are definite strengths and weakness. The strengths include a fairly well written description of the methodology and the careful comparison of the authors program vs other established image analysis programs/approaches that are being currently used in the field. Weaknesses include limited impact - the manuscript describes an image analysis program that is better than other existing programs, but it's not clear to me how significant this improvement means to the field. Additionally, the manuscript is a methods-only paper, not addressing any biological question or demonstrating how the authors software could shed light on this question where other programs fall short.

Reply: The reviewer is correct that our manuscript is a methods-only paper. We believe biological hypothesis testing would unnecessarily lengthen the paper and distract from the core methodological innovation. We believe that *BCM3D* will be useful for researchers in the field because:

- *BCM3D* is generally applicable. We demonstrated automated cell segmentation AND morphological cell classification capabilities of *BCM3D* using spherical and rod-shaped cells, as well as cells labeled with cytosolic fluorescent proteins, cell membrane staining, and mixtures of the two. *BCM3D* is novel and impactful, because, to our knowledge, there are no other cell segmentation approaches that can provide these capabilities. These capabilities are particularly important when multispecies biofilms are imaged in a single color or when different labeling strategies are used to provide complementary information about the cells being imaged, such as cell shapes and gene activation.

(Lines 615ff): *“The ability to separate different cell morphologies is important for the study of multispecies biofilms, where interspecies cooperation and competition dictate population-level outcomes⁸⁻¹⁶. Separation of differentially labeled cells is also important for the study of gene activation in response to cell-to-cell signaling¹⁷. Expression of cytosolic fluorescent proteins by transcriptional reporter strains is a widely-used technique to visualize activation of a specific gene or genetic pathway in living cells. Such genetic labeling approaches can be complemented by chemical labeling approaches, e.g. using membrane intercalating chemical dyes that help visualize cells non-specifically or environmentally-sensitive membrane dyes that provide physiological information, including membrane composition^{18,19}, membrane organization and integrity²⁰⁻²², and membrane potential^{23,24}. Chemical and genetic labeling approaches are traditionally implemented in two different color channels. However, there are important drawbacks to using multiple colors. First and foremost, the amount of excitation light delivered is increased to excite differently colored fluorophores, raising phototoxicity and photobleaching concerns. Second, it takes N times as long to acquire N -color images (unless different color channels can be acquired simultaneously), making it challenging to achieve high temporal sampling in time-lapse acquisition. For these reasons,*

methods that extract complementary physiological information from a single-color image are preferable.”

- BCM3D can be adapted even by non-experts in digital signal processing. As mentioned by Reviewer 1, CNN-based processing is an “elegant, end-to-end trained method”, for which limited optimization is required. Our *LCuts*-based post-processing module distinguishes itself from other mathematical signal processing steps that rely on hard-to-understand and sample-specific parameter optimization. In contrast, *LCuts* post-processing requires only four adjustable parameters that can be selected based on *a priori* biological knowledge. The performance of *LCuts* is robust to changes in these parameters (see our response to **Reviewer Comment 1.1**). We believe that the *BCM3D* workflow will be rapidly adapted by the field, because extensive parameter optimization is not needed.

(Lines 364ff): *“Furthermore, to limit the need for optimization of postprocessing routines, the four adjustable parameters used in LCuts, namely cell diameter d , the cell length L , and the decay parameters σ_D and σ_T are chosen based on a priori knowledge about the bacterial cells under investigation. We found that the performance of LCuts is not sensitive to the particular values of d , L , σ_D and σ_T as long as they are consistent with the imaged bacterial cell sizes and shapes (Figure S5).”*

- The presented systematic comparison of BCM3D performance to other algorithms in the field fills a critical knowledge gap. We systematically evaluate cell morphometry and cell counting accuracies achieved by available algorithms using simulated biofilm images with varying cell densities and SBRs similar to those encountered in experimental data.

(Lines 423ff): *“The cell shapes to be segmented are densely packed and barely resolvable even with the highest resolution optical microscopes. Additionally, living biofilms in fluorescence microscopes can only be imaged with low laser intensities to ameliorate phototoxicity and photobleaching concerns. Unfortunately, low intensity fluorescence excitation also reduces the SBR in the acquired images. So far, it remains unclear to what extent single-cell segmentation approaches can accurately identify and delineate cell shapes in bacterial biofilm images obtained under low intensity illumination conditions.”*

- BCM3D provides state-of-the-art performance, especially for low SBR/high cell density datasets that are particularly challenging for other image processing methods. Low SBR conditions are commonly encountered when researchers seek to ameliorate phototoxicity and photobleaching during imaging.

(Lines 110ff): *“BCM3D also achieves higher segmentation accuracy on experimental 3D biofilm data than Cellpose⁶, a state-of-the-art, CNN-based, generalist algorithm for cell segmentation and the algorithm used by Hartmann et al.², a specialized algorithm designed for bacterial cell segmentation based on traditional mathematical image processing.”*

and

(Lines 456ff): *“Overall, the cell counting accuracies obtained by BCM3D are higher than other methods and remain higher even for IoU matching thresholds larger than 0.5, indicating that cell shapes are more accurately estimated by the in silico-trained CNNs.”*

In summary, the new capabilities provided by *BCM3D* when combined with a non-invasive imaging technologies, like lattice light sheet microscopy, will enable accurate segmentation of individual cells in crowded environments and automatic assignments of individual cells to specific cell populations (Line 114ff). We believe that *BCM3D* will help “launch a new era for bacterial biofilm research, in which the emergent properties of microbial populations can be studied in terms of the fully-resolved behavioral phenotypes of individual cells.” (Line 116ff)

References

1. Hartmann, R., Jeckel, H., Jelli, E., Singh, P.K., Vaidya, S., Bayer, M., Vidakovic, L., Díaz-Pascual, F., Fong, J.C.N., Dragoš, A., Besharova, O., Nadell, C.D., Sourjik, V., Kovács, Á.T., Yildiz, F.H. & Drescher, K. BiofilmQ, a software tool for quantitative image analysis of microbial biofilm communities. *BioRxiv* 735423 (2019).
2. Hartmann, R., Singh, P.K., Pearce, P., Mok, R., Song, B., Díaz-Pascual, F., Dunkel, J. & Drescher, K. Emergence of three-dimensional order and structure in growing biofilms. *Nature Physics* **15**, 251-256 (2019).
3. Berg, S., Kutra, D., Kroeger, T., Straehle, C.N., Kausler, B.X., Haubold, C., Schiegg, M., Ales, J., Beier, T., Rudy, M., Eren, K., Cervantes, J.I., Xu, B., Beuttenmueller, F., Wolny, A., Zhang, C., Koethe, U., Hamprecht, F.A. & Kreshuk, A. ilastik: interactive machine learning for (bio)image analysis. *Nature Methods* (2019).
4. McQuin, C., Goodman, A., Chernyshev, V., Kametsky, L., Cimini, B.A., Karhohs, K.W., Doan, M., Ding, L., Rafelski, S.M., Thirstrup, D., Wiegraebe, W., Singh, S., Becker, T., Caicedo, J.C. & Carpenter, A.E. CellProfiler 3.0: Next-generation image processing for biology. *PLoS Biol* **16**, e2005970 (2018).
5. Weigert, M., Schmidt, U., Haase, R., Sugawara, K. & Myers, G. Star-convex Polyhedra for 3D Object Detection and Segmentation in Microscopy. in *2020 IEEE Winter Conference on Applications of Computer Vision (WACV)* 3655-3662 (2020).
6. Stringer, C., Wang, T., Michaelos, M. & Pachitariu, M. Cellpose: a generalist algorithm for cellular segmentation. *bioRxiv*, 2020.2002.2002.931238 (2020).
7. Jaccard, P. The distribution of the flora in the alpine zone. *New Phytologist* **11**, 37-50 (1912).
8. Liu, J., Prindle, A., Humphries, J., Gabalda-Sagarra, M., Asally, M., Lee, D.Y., Ly, S., Garcia-Ojalvo, J. & Suel, G.M. Metabolic co-dependence gives rise to collective oscillations within biofilms. *Nature* **523**, 550-554 (2015).
9. Prindle, A., Liu, J., Asally, M., Ly, S., Garcia-Ojalvo, J. & Suel, G.M. Ion channels enable electrical communication in bacterial communities. *Nature* **527**, 59-63 (2015).
10. Humphries, J., Xiong, L., Liu, J., Prindle, A., Yuan, F., Arjes, H.A., Tsimring, L. & Suel, G.M. Species-Independent Attraction to Biofilms through Electrical Signaling. *Cell* **168**, 200-209 e212 (2017).
11. Liu, J., Martinez-Corral, R., Prindle, A., Lee, D.-y.D., Larkin, J., Gabalda-Sagarra, M., Garcia-Ojalvo, J. & Suel, G.M. Coupling between distant biofilms and emergence of nutrient time-sharing. *Science* **356**, 638-642 (2017).

12. Mitri, S. & Foster, K.R. The genotypic view of social interactions in microbial communities. *Annu Rev Genet* **47**, 247-273 (2013).
13. Drescher, K., Nadell, C.D., Stone, H.A., Wingreen, N.S. & Bassler, B.L. Solutions to the public goods dilemma in bacterial biofilms. *Current biology : CB* **24**, 50-55 (2014).
14. Persat, A., Nadell, C.D., Kim, M.K., Ingremeau, F., Siryaporn, A., Drescher, K., Wingreen, N.S., Bassler, B.L., Gitai, Z. & Stone, H.A. The mechanical world of bacteria. *Cell* **161**, 988-997 (2015).
15. Papenfort, K. & Bassler, B.L. Quorum sensing signal-response systems in Gram-negative bacteria. *Nat Rev Microbiol* **14**, 576-588 (2016).
16. Nadell, C.D., Drescher, K. & Foster, K.R. Spatial structure, cooperation and competition in biofilms. *Nat Rev Microbiol* **14**, 589-600 (2016).
17. Kroos, L. Highly Signal-Responsive Gene Regulatory Network Governing Myxococcus Development. *Trends Genet* **33**, 3-15 (2017).
18. Moon, S., Yan, R., Kenny, S.J., Shyu, Y., Xiang, L., Li, W. & Xu, K. Spectrally resolved, functional super-resolution microscopy reveals nanoscale compositional heterogeneity in live-cell membranes. *Journal of the American Chemical Society* **139**, 10944-10947 (2017).
19. Bramkamp, M. & Lopez, D. Exploring the existence of lipid rafts in bacteria. *Microbiol. Mol. Biol. Rev.* **79**, 81-100 (2015).
20. Zou, S.B., Hersch, S.J., Roy, H., Wiggers, J.B., Leung, A.S., Buranyi, S., Xie, J.L., Dare, K., Ibba, M. & Navarre, W.W. Loss of elongation factor P disrupts bacterial outer membrane integrity. *Journal of bacteriology* **194**, 413-425 (2012).
21. Gonelimali, F.D., Lin, J., Miao, W., Xuan, J., Charles, F., Chen, M. & Hatab, S.R. Antimicrobial properties and mechanism of action of some plant extracts against food pathogens and spoilage microorganisms. *Frontiers in microbiology* **9**, 1639 (2018).
22. Parasassi, T., De Stasio, G., d'Ubaldo, A. & Gratton, E. Phase fluctuation in phospholipid membranes revealed by Laurdan fluorescence. *Biophysical journal* **57**, 1179-1186 (1990).
23. Strahl, H. & Hamoen, L.W. Membrane potential is important for bacterial cell division. *Proceedings of the National Academy of Sciences* **107**, 12281-12286 (2010).
24. Prindle, A., Liu, J., Asally, M., Ly, S., Garcia-Ojalvo, J. & Süel, G.M. Ion channels enable electrical communication in bacterial communities. *Nature* **527**, 59-63 (2015).

Reviewers' Comments:

Reviewer #1:

Remarks to the Author:

The authors have addressed my concerns and suggestions with new analyses, which I appreciate seeing. As before, I find the approach well justified, well executed and successful, and I don't have any remaining caveats to add to these statements. - Marius Pachitariu

Reviewer #2:

Remarks to the Author:

The authors have answered all my questions. I have no further comments.

Response to Reviewer Comments (original submission)

Note to all reviewers:

After we posted our manuscript on bioRxiv, we received some feedback and suggestions from Prof. Knut Drescher, the corresponding author of *BiofilmQ*¹. He pointed out that *BiofilmQ* was not specifically designed for single-cell segmentation and suggested that we instead compare our results to those obtained by an algorithm developed by Hartmann *et al.*². The Hartmann *et al.* algorithm is a highly-refined bacterial cell segmentation algorithm based on traditional mathematical image processing routines. Because the source code of the Hartmann *et al.* algorithm is not yet publicly available, Prof. Drescher offered to run the algorithm on simulated and experimental images that we would provide to his lab. We found that the Hartmann *et al.* algorithm indeed outperformed *BiofilmQ* and other algorithms based on traditional mathematical image processing and it can therefore be regarded as the state-of-the-art algorithm in this category. We have therefore replaced any references pertaining to and any results obtained with *BiofilmQ* with those of the Hartmann *et al.* algorithm. The Drescher lab is now acknowledged for their contribution in the acknowledgement section of the revised manuscript. This change does not affect the conclusions we draw, but instead strengthens the significance of *BCM3D* as a more accurate and versatile segmentation approach.

Reviewer #1:

This paper describes a computational method for achieving good segmentation performance on bacterial biofilms composed of cells of various shapes. The approach is well justified, well executed and highly successful. It follows a modern approach of 1) simulating lots of training data + 2) training a deep CNN + 3) post-processing the results of the CNN. The simulation can be adjusted by the user to match the parameters of the real data the user is interested in segmenting, which gives this method more flexibility than a lot of other CNN-based approaches. The approach is shown to outperform older, non-CNN based methods by a wide margin. The authors even include comparisons to the very recently released Cellpose package, showing that their method consistently works as well as or better for the specific types of images used in this paper. Generally speaking, I like everything about this paper.

Reply: We thank the reviewer for this positive overall assessment.

Comment 1.1

The thing I like least, that I would like to see some changes to in a revision, relates to the postprocessing workflow that follows the pixel predictions from the deep CNNs. This is always a point of concern for CNN-based methods, because the CNN is a relatively elegant, end-to-end trained method, while the post-processing tends to be idiosyncratic and manually optimized to the specific data we are seeing. It is difficult to know if this approach will work on other types of data, mainly because of this post-processing step. I think it's important to clearly state if all the hyper-parameters in the post-processing were chosen on validation data, as opposed to on the test data on which results are reported. If they were optimized on the test data, then the optimization needs

to be re-done on a separate validation dataset and the best set of hyper-parameters be picked automatically with grid searches, not by a human with subjective biases.

Reply:

To limit the need for manual (idiosyncratic) optimization of postprocessing routines, the three adjustable parameters used in *LCuts* are NOT determined through data-driven optimization. Instead, these parameters are chosen based on a priori knowledge regarding the bacterial cells under investigation. These parameters are the cell diameter d , the cell length L , and the distance decay parameter σ_D . The cell diameters and lengths can be readily estimated based on images of dispersed bacterial cells and the distributions of these parameters over a given cell population is therefore well-defined for many bacterial species. If they are not available from literature sources, then the distributions can be readily obtained through a separate experiment in which biofilm cells are dispersed and then imaged in isolation. Such experiments are routine in microbiology. Here, we use the median value of the cell diameter to select d . This constrains the radii of the inscribed spheres to the interval $[d/2 - 0.2, d/2 + 0.2]$ μm for medial axis extraction (postprocessing step 2). Similarly, for linear clustering (postprocessing step 3), we constrain the cell lengths to be less than the maximum expected cell length L in the population. The distance decay parameter σ_D was chosen as the maximum cell radius.

We have included an additional section in the manuscript starting on Lines 364:

*“Furthermore, to limit the need for optimization of postprocessing routines, the four adjustable parameters used in *LCuts*, namely cell diameter d , the cell length L , and the decay parameters σ_D and σ_T are chosen based on a priori knowledge about the bacterial cells under investigation. We found that the performance of *LCuts* is not sensitive to the particular values of d , L , σ_D and σ_T as long as they are consistent with the imaged bacterial cell sizes and shapes (**Figure S4**). Identification of single cells provided by *LCuts* alleviates under-segmentation errors of the CNN-based segmentation.”*

Figure S4 validates our parameter selection by showing the results of a grid search on 20 randomly chosen, simulated datasets of low SBR and/or high cell density, for which post-processing is required to segment single cells. The simulated cells have a median diameter of 0.8 μm and vary in length from 2 to 6 μm .

Figure S4. Validation of parameter selection for *LCuts* postprocessing by grid search. Shown is the cell counting accuracy averaged over 20 randomly chosen, simulated datasets of low SBR and/or high cell density, for which post-processing is required. **(a)** Average cell counting accuracy as a function of cell diameter $d \in [0.4, 1.2] \mu\text{m}$ and cell length $L \in [2, 9] \mu\text{m}$ at a fixed $\sigma_D = 0.5 \mu\text{m}$ and $\sigma_T = 0.2$. **(b)** Average cell counting accuracy as a function of $\sigma_D \in [0.1, 0.8] \mu\text{m}$, and $\sigma_T \in [0.05, 0.6]$ with fixed $(d, L) = (0.8, 4.5) \mu\text{m}$. The cell counting accuracy is largely unaffected by variations in d , L , σ_D and σ_T and robustly remains above 70% for biologically reasonable parameter values, such as $d \sim 0.8 \mu\text{m}$, cell length $L \sim 6 \mu\text{m}$, for *E.coli*-like cell shapes. We also choose $\sigma_D = d/2$ and $\sigma_T = 0.2$, so that edges between nodes separated by more than a cells radius or with relative angles $>30^\circ$ are weighted down.

Comment 1.2

The next part of my concerns relates to the reproducibility of this post-processing step. It doesn't help that it's in Matlab, which is closed source and unavailable to a lot of people. The rest of the code used is open source, so I recommend the authors consider rewriting their post-processing step in Python. Even better, why not use an existing package that can do the post-processing for you? The first things that come to mind are *ilastik* and *CellProfiler*, but there are many others too. A watershed based algorithm seems like it could deal pretty well with the merge problems generated by the soft pixel predictions of the CNN. Perhaps I am wrong, and the *L-Cuts* algorithm is really needed, but in that case I'd still like to see it shown in the paper that a simpler post-processing method was insufficient.

Reply:

We agree with the reviewer that newly developed algorithms should be available in an open source format. We would like to point out that, while MATLAB itself is a commercial closed source platform, our MATLAB code is open source and executable on Octave, which is an open-source platform. We are however committed to providing a Python version of our code in our Github repository in the future to ensure broader adaption in the field.

We now include a new **Figure S7**) that compares *LCuts* to other open source post-processing algorithms, namely the hysteresis thresholding-based algorithm of *Ilstik* and

the watershed-based pipeline of *CellProfiler*. For the test data, we again chose the same 20 simulated datasets used in **Figure S4**. The results show that the hysteresis thresholding-based algorithm improves the cell counting accuracy by less than 6% on average. On the other hand, the watershed-based pipeline used by *CellProfiler* decreases the cell counting accuracy in many cases, which is primarily due to oversegmentation. Among the three methods tested, *LCuts* provides the highest improvement in cell counting accuracy (>12% on average for IoU matching thresholds less than 0.6). These data indicate that simpler post-processing methods commonly used by researchers do not provide the performance boost of *LCuts*. Therefore, the application of *LCuts* is indeed necessary. **Figure S7** is referenced on Line 518 of the manuscript:

“LCuts thus provides an important benefit in improving the cell counting accuracy to an extent not achieved by currently available thresholding- or watershed-based post-processing algorithms (Figure S7).”

Figure S7. Comparison of *LCuts* to commonly used image post-processing methods. Shown is the cell counting accuracy averaged over 20 randomly chosen, simulated datasets of low SBR and/or high cell density, for which post-processing is required. The hysteresis thresholding-based algorithm of *Ilastik*³ improves the cell counting accuracy by less than 6% on average for IoU matching thresholds less than 0.6. On the other hand, the watershed-based pipeline used by *CellProfiler* 3.0⁴ provides negligible improvements and even decreases the average cell counting accuracy in many cases. This decrease is primarily due to oversegmentation. Among the three methods tested, *LCuts* provides the highest improvement in cell counting accuracy (>12% on average for IoU matching thresholds less than 0.6).

Reviewer #2:

The article by Zhang et al. presents a new segmentation workflow consisting of a convolutional neural network whose results are post-processed by a graph-based data clustering method developed by the same group. The authors show that they can segment bacteria in biofilms with the segmentation accuracy depending on density and signal to background ratio (SBR). What makes the article interesting is not only the performance but also that they used simulations to train the CNN and thus can circumvent the laborious collection of training sets.

Reply: We thank the reviewer for this positive overall assessment.

Comment 2.1

The authors mention depth dependent aberrations only in so far as stating that they don't consider them. But a comparison of simulations and experiments could show how strong that influence is. Does segmentation accuracy change with biofilm depth? Presumably yes if the SBR changes.

Reply: Yes, the segmentation accuracy does change with biofilm depth, because the SBR decreases. The Figure below shows a decrease in SBR and an associated decrease in cell segmentation accuracy as a function of imaging depth. The images are of a thick 3D biofilm of spherical *Myxococcus xanthus* spores. *M. xanthus* spores are highly scattering due to their polysaccharide spore coats, so the dependence of SBR on imaging depth is very pronounced. We chose not to include this Figure in the Supporting Information, because a thorough characterization of segmentation performance as a function of imaging depth is species-specific. Instead, we chose to characterize segmentation performance as a function of SBR which is a more general metric.

Comment 2.2

375/6: An IoU of 0.5 seems low? It would be instructive if the authors could produce a distribution of IoU values to justify that cutoff and indicate how strongly that cutoff influences the correct segmentation of cells.

Reply: The choice of IoU matching threshold is arbitrary. A common practice in the field is to choose an IoU matching threshold of 0.5 when comparing single values of cell counting accuracy^{5,6}. We follow this convention in our manuscript as well. The more important point however is that cell counting accuracy is a function of IoU matching threshold and therefore this functional dependence is shown in many of our Figures, namely **Figure 3ghi**, **Figure 5abc**, **Figure 6d**, **Figure 7cd**, and **Figure S6**.

We have edited the text to clarify this point (Lines 380):

“We quantified segmentation accuracy both at the cell-level (object counting) and at the voxel-level (cell shape estimation). To quantify the cell-level segmentation accuracy, we designated segmented objects as true positive (TP) if their voxel overlap with the ground truth or the manual annotation resulted in an IoU value larger than a particular IoU matching threshold. This criterion ensures one-to-one matching. A threshold of 0.5 is typically chosen when reporting single cell counting accuracy values^{5,6}. We follow this convention here.”

Comment 2.3

The authors already show graphs in Figs S2 and S5 to address this issue. But it seems to reach a high confidence level one would need an IoU significantly higher than 0.5? And why does the IoU value drop at high confidence thresholds? In other words, a confidence threshold of 0.65 and 0.99 have the same IoU. So the IoU value cannot capture the quality of segmentation unambiguously?

Reply: We thank the reviewer for highlighting this possible point of confusion. Indeed, there are two different thresholds used in our paper.

The first threshold, the confidence value threshold, is used to assign voxels from the CNN output (containing confidence values ranging from 0 to 1) to either ‘cell interior’ or ‘background’ classes. This step will result in a binary image, which is then used for further processing, as explained in the manuscript (Lines 270ff and 287ff). The selection of the confidence value threshold is based on the data shown in **Figure S2**, which shows the voxel-level segmentation accuracy of the entire image as a function of the confidence value threshold. The reason why a confidence value threshold of 0.65 and 0.99 give a similar voxel-level segmentation accuracy is as follows: Too low confidence thresholds, such as 0.65, produce many connected objects as well as many artificial objects. This reduces the voxel-level segmentation accuracy. Too high confidence value thresholds, such as 0.99, filter out too many (true positive) cells. This also reduces the voxel-level segmentation accuracy. The optimal voxel-level segmentation accuracy is consistently obtained using confidence value thresholds between 0.88 and 0.94. Throughout this work, we use 0.88 for cells labeled with intracellular fluorophores and 0.94 for cells labeled with membrane-localized fluorophores.

We have edited the sentences on Line 279ff to clarify this point:

“The threshold value to segment individual cell objects based on the ‘cell interior’ confidence map was determined by plotting the overall voxel-level segmentation accuracy, quantified as the Intersection-over-Union value (IoU value, aka Jaccard index²) versus the confidence value thresholds (Figure S2). Optimal voxel-level segmentation accuracies were consistently obtained using confidence thresholds between 0.88 and 0.94. Throughout this work, we use 0.94 for cells labeled with intracellular fluorophores and 0.88 for cells labeled with membrane-localized fluorophores.”

We also changed the *x*-axis label of **Figure S2** to “Confidence value thresholds” and the *y*-axis label to “Voxel-level segmentation accuracy” to avoid confusion with the second threshold used in our paper, the IoU matching threshold. The IoU matching threshold is used to quantify the cell counting accuracy, as explained in section **Performance Evaluation** (Lines 379ff).

Comment 2.4

The fact that voxel IoU and cell number accuracy is not the same, points to problems with shape determination. Perhaps authors can discuss this problem in more detail and potentially use erosion/dilation algorithms to better adapt shapes? At the moment the authors discuss this issue only in relation to other algorithms.

Reply: The reviewer is correct that voxel-level segmentation accuracy and cell counting accuracy are not the same. As explained in **Performance Evaluation** (Lines 379ff), these two metrics quantify segmentation accuracy respectively at the voxel-level (cell shape estimation) and at the cell-level (object counting). We disagree, however, that the results point to problems with shape determination. Cell shapes are estimated accurately if the voxel-level segmentation accuracy is high. Single cells are accurately segmented as individual objects, if the cell counting accuracy is high. Both of these metrics need to be high to accurately segment single cell objects AND their shapes, as indicated by the dashed arrows in Figures 3ghi.

We also note that dilation is used in our workflow after thresholding the CNN confidence maps and object identification to increase cell size, as explained on Lines 277ff.

Comment 2.5

Could the authors provide difference maps between a) simulation GT and segmentation, b) between manual annotation of simulations and automatic segmentation? This would show on one side the quality of the segmentation but also show how manual segmentation is comparing to automatic segmentation when done on the same ground truth (see lines 550-561 where the authors already hint at that problem). Also in Fig. S7 such a difference map would be very helpful.

Reply: We thank the reviewer for the suggestion to visually assess the quality of our segmentation and manual annotation. Indeed, the segmentation accuracy metrics are lower, when they are evaluated with respect to the manual annotation result instead of the ground

truth. As the reviewer noticed, we already hinted at this problem in the manuscript (now Lines 565-573) To more thoroughly quantify this effect, the revised manuscript now includes a new **Figure S9** (shown below). The data in **Figure S9** substantiates our claim that manual annotations don't determine cell shapes as reliably as the ground truth. (Please also see our Reply to **Comment 2.6** below).

In addition to **Figure S9**, we have also made a slice-by-slice video showing differences in segmentation and ground truth for the mixed-shape biofilm shown in **Figure S11**.

Figure S9 (a) Fluorescence image slice of a 3D simulated biofilm and (b) the corresponding ground truth (GT). (c) The fluorescence image slice shown in (a) masked by its corresponding GT shown in (b). The fluorescence is not completely masked because of the diffraction-limited resolution of light microscopy. (d) Fluorescence image slice of the same simulated biofilm masked by the *BCM3D* segmentation result. (e) Absolute value of the difference image between the GT and the *BCM3D* segmentation result. White pixels indicate regions where the two masks do not agree. (f) Absolute value of the difference

image between a manual annotated mask (from researcher 3) and the *BCM3D* segmentation result. **(g)** Fluorescence image slice of the same simulated biofilm masked by the manual annotation result. Researcher 3 chose to draw larger cell boundaries to mask more of the fluorescence intensity. **(h)** Absolute value of the difference image between the GT and the manually annotated mask. White pixels indicate regions where the two masks do not agree. **(I)** Segmentation accuracy achieved by manual annotation performed by three different researchers. Segmentation accuracy is parameterized in terms of cell counting accuracy (*y* axis) and IoU matching threshold (*x* axis, a measure of cell shape estimation accuracy). Curves approaching the upper right-hand corner indicate higher overall segmentation accuracy with respect to the ground truth. While IoU matching thresholds <0.3 yield good cell counting accuracies, the cell counting accuracy sharply decreases for IoU matching thresholds >0.3 , because manually annotated cell shapes differ from the ground truth cell shapes.

Comment 2.6

Fig. 5: there are strong differences between manual and *BCM3D* segmentation. Again a comparison on simulations might be good as a ground truth for comparison would be available. But it is also clear that the manual segmentation leads in general to smaller cells. Difference maps to the raw data would be very interesting.

Reply: The reviewer is correct that there are differences between manual annotation and *BCM3D* segmentation. These differences are now highlighted in the new **Figure S9**, (see response to previous comment). (Unfortunately), manual annotation masks are the only references to which segmentation results of experimental images can be compared. The manual annotation and *BCM3D* segmentation mismatch is also reflected in the fact that the curves in **Figure 5abc** and **Figure S9i** do not approach the upper right hand corner. We have edited the manuscript and added a new **Figure S10** to better highlight and clarify this issue (Lines 565ff):

*“We attribute the more rapid drop-off of the cell counting accuracy as a function of increasing IoU matching threshold in **Figure 5** to the following factors. First, human annotation of experimentally acquired biofilm images differs from the ground truth segmentation masks that are available for simulated data (**Figure S9**). The shape mismatches between algorithm segmented and manually annotated cell shapes (**Figure S9** and **S10**) lead to a global lowering of voxel-level segmentation accuracy and thus a more rapid drop-off of the cell counting accuracy as a function of increasing IoU matching threshold. Because bacterial cell shapes are not accurately captured by manual annotation (**Figures S9**), cell counting accuracies referenced to manual annotations should be compared only at low IoU matching thresholds (0.1-0.3), as also noted previously⁵.”*

Figure S10 (a) Fluorescence image slice of the 3D *E. coli* biofilm shown in **Figure 5c** masked by the *BCM3D* segmentation result. (b) Fluorescence image slice of the 3D *E. coli* biofilm shown in **Figure 5c** masked by manual annotation. (c) Absolute value of the difference image between manual annotation and *BCM3D* segmentation mask. Black pixels indicate image regions where the two masks agree and white pixels indicate image regions where the two masks differ.

Comment 2.7

528: is it clear when *Cellpose* or *BCM3D* is better? On average better is not a strong statement.

Reply: We thank the reviewer for pointing out the ambiguity related to the term “on average”. The numerically quantified performance of *BCM3D* is slightly better than other state-of-the-art methods for this particular *E. coli* biofilm. However, the qualitative performance is substantially better than other state-of-the-art methods. We have edited the manuscript text to clarify this point (Lines 541ff):

“However, mathematical post-processing of the CNN outputs by LCuts corrects some of these errors, so that the integrated BCM3D workflow achieves improved results compared to Cellpose and Hartmann et al. at each of the indicated time points. Visual inspection of the segmentation results is also informative. Cellpose accurately segments individual cells in low density regions, but suffers from oversegmentation errors in high density biofilm regions (Figure S8e). The Hartmann et al. algorithm provides reasonable estimates of cell positions in low and high density biofilm regions, but again struggles with cell shape estimation (Figure S8d, see also Figure 3g-i). On the other hand, the integrated BCM3D workflow (CNN + LCuts) produces biologically reasonable cell shapes regardless of cell density (Figure 5).”

We also note that, the numerically quantified performance of *BCM3D* is substantially better than other state-of-the-art methods for the simulated dataset in **Figure 3** and **4** as well as the membrane-labeled *M. xanthus* cells in **Figure 6**. We have edited the manuscript text to clarify this point as well (Lines 581ff):

“To demonstrate that BCM3D can achieve similarly high segmentation accuracies for membrane-stained cells in different cellular arrangements, we analyzed a small patch of a M. xanthus biofilm, which was stained with the membrane intercalating dye FM4-64 (Figure 6a). In contrast to E. coli biofilms, the submerged M. xanthus biofilm imaged here features cells in a mesh-like arrangement with close cell-to-

cell contacts, which presents a unique challenge for 3D single-cell segmentation. To obtain reference data for 3D segmentation accuracy determination, we manually annotated each xy, xz, and yz slice of an entire 3D image stack (Figure 6b). When referenced to this 3D manual segmentation result, BCM3D (Figure 6c) produced cell counting accuracies above 70% at low (0.1-0.3) IoU matching thresholds, whereas segmentation results obtained by conventional image processing (Hartmann et al.) and by generalist CNN-processing (Cellpose) produced cell counting accuracies <50% in the same IoU matching threshold region (Figure 6d). We note however that neither Cellpose nor the Hartmann et al. algorithm were specifically optimized/ designed for segmenting membrane-stained cells. Indeed, the performance of Cellpose on this type of biofilm architecture is inferior to the results achieved using the in silico-trained CNNs of BCM3D alone (without using LCuts post-processing). One reason might be that the pre-trained, generalist Cellpose model has not been trained sufficiently on long, thin, and highly interlaced rod-shaped cells, such as those contained in a M. xanthus biofilm.”

Comment 2.8

617: independent of population fraction? But that must be density dependent? Please clarify.

Reply: The reviewer is correct that segmentation accuracy is cell density-dependent. That fact is made clear by the data shown in **Figure 3** and **Figure 4**. For the simulation of the two-population biofilm images, we chose a SBR (2.56), for which the segmentation accuracy varies little with cell density (unless cell density is extremely high >60%).

The cell densities vary slightly for the biofilm data used in **Figure 7a**, because cell position are first simulated using rod shaped cells. A predetermined fraction of rod-shaped cells is then replaced with spherical cells. These replacements generate unoccupied volumes in the biofilm. The reason for this ad-hoc approach is that *CellModeller* and similar freely available individual-based models do not allow for simulations of mixed cell shapes. We have corrected the manuscript text to clarify this point (Lines 241 ff):

“Cell arrangements were first simulated using rod shaped cells and then a fraction of rod-shaped cells is replaced with spherical cells.”

We have now added a secondary axis label to **Figure 7ab** to show the average cell densities at each population mixing ratio. We also specifically state the simulated cell density of the biofilm shown in **Figure S11**.

Reviewer #3:

In this manuscript the authors introduce a biofilm image analysis tool that combines machine learning and improved image analysis to resolve individual cells within biofilms.

Comment 3.1

There are definite strengths and weakness. The strengths include a fairly well written description of the methodology and the careful comparison of the authors program vs other established image analysis programs/approaches that are being currently used in the field. Weaknesses include limited impact - the manuscript describes an image analysis program that is better than other existing programs, but it's not clear to me how significant this improvement means to the field. Additionally, the manuscript is a methods-only paper, not addressing any biological question or demonstrating how the authors software could shed light on this question where other programs fall short.

Reply: The reviewer is correct that our manuscript is a methods-only paper. We believe biological hypothesis testing would unnecessarily lengthen the paper and distract from the core methodological innovation. We believe that *BCM3D* will be useful for researchers in the field because:

- *BCM3D* is generally applicable. We demonstrated automated cell segmentation AND morphological cell classification capabilities of *BCM3D* using spherical and rod-shaped cells, as well as cells labeled with cytosolic fluorescent proteins, cell membrane staining, and mixtures of the two. *BCM3D* is novel and impactful, because, to our knowledge, there are no other cell segmentation approaches that can provide these capabilities. These capabilities are particularly important when multispecies biofilms are imaged in a single color or when different labeling strategies are used to provide complementary information about the cells being imaged, such as cell shapes and gene activation.

(Lines 615ff): *“The ability to separate different cell morphologies is important for the study of multispecies biofilms, where interspecies cooperation and competition dictate population-level outcomes⁸⁻¹⁶. Separation of differentially labeled cells is also important for the study of gene activation in response to cell-to-cell signaling¹⁷. Expression of cytosolic fluorescent proteins by transcriptional reporter strains is a widely-used technique to visualize activation of a specific gene or genetic pathway in living cells. Such genetic labeling approaches can be complemented by chemical labeling approaches, e.g. using membrane intercalating chemical dyes that help visualize cells non-specifically or environmentally-sensitive membrane dyes that provide physiological information, including membrane composition^{18,19}, membrane organization and integrity²⁰⁻²², and membrane potential^{23,24}. Chemical and genetic labeling approaches are traditionally implemented in two different color channels. However, there are important drawbacks to using multiple colors. First and foremost, the amount of excitation light delivered is increased to excite differently colored fluorophores, raising phototoxicity and photobleaching concerns. Second, it takes N times as long to acquire N -color images (unless different color channels can be acquired simultaneously), making it challenging to achieve high temporal sampling in time-lapse acquisition. For these reasons,*

methods that extract complementary physiological information from a single-color image are preferable.”

- BCM3D can be adapted even by non-experts in digital signal processing. As mentioned by Reviewer 1, CNN-based processing is an “elegant, end-to-end trained method”, for which limited optimization is required. Our *LCuts*-based post-processing module distinguishes itself from other mathematical signal processing steps that rely on hard-to-understand and sample-specific parameter optimization. In contrast, *LCuts* post-processing requires only four adjustable parameters that can be selected based on *a priori* biological knowledge. The performance of *LCuts* is robust to changes in these parameters (see our response to **Reviewer Comment 1.1**). We believe that the *BCM3D* workflow will be rapidly adapted by the field, because extensive parameter optimization is not needed.

(Lines 364ff): *“Furthermore, to limit the need for optimization of postprocessing routines, the four adjustable parameters used in LCuts, namely cell diameter d , the cell length L , and the decay parameters σ_D and σ_T are chosen based on a priori knowledge about the bacterial cells under investigation. We found that the performance of LCuts is not sensitive to the particular values of d , L , σ_D and σ_T as long as they are consistent with the imaged bacterial cell sizes and shapes (Figure S5).”*

- The presented systematic comparison of BCM3D performance to other algorithms in the field fills a critical knowledge gap. We systematically evaluate cell morphometry and cell counting accuracies achieved by available algorithms using simulated biofilm images with varying cell densities and SBRs similar to those encountered in experimental data.

(Lines 423ff): *“The cell shapes to be segmented are densely packed and barely resolvable even with the highest resolution optical microscopes. Additionally, living biofilms in fluorescence microscopes can only be imaged with low laser intensities to ameliorate phototoxicity and photobleaching concerns. Unfortunately, low intensity fluorescence excitation also reduces the SBR in the acquired images. So far, it remains unclear to what extent single-cell segmentation approaches can accurately identify and delineate cell shapes in bacterial biofilm images obtained under low intensity illumination conditions.”*

- BCM3D provides state-of-the-art performance, especially for low SBR/high cell density datasets that are particularly challenging for other image processing methods. Low SBR conditions are commonly encountered when researchers seek to ameliorate phototoxicity and photobleaching during imaging.

(Lines 110ff): *“BCM3D also achieves higher segmentation accuracy on experimental 3D biofilm data than Cellpose⁶, a state-of-the-art, CNN-based, generalist algorithm for cell segmentation and the algorithm used by Hartmann et al.², a specialized algorithm designed for bacterial cell segmentation based on traditional mathematical image processing.”*

and

(Lines 456ff): *“Overall, the cell counting accuracies obtained by BCM3D are higher than other methods and remain higher even for IoU matching thresholds larger than 0.5, indicating that cell shapes are more accurately estimated by the in silico-trained CNNs.”*

In summary, the new capabilities provided by *BCM3D* when combined with a non-invasive imaging technologies, like lattice light sheet microscopy, will enable accurate segmentation of individual cells in crowded environments and automatic assignments of individual cells to specific cell populations (Line 114ff). We believe that *BCM3D* will help “launch a new era for bacterial biofilm research, in which the emergent properties of microbial populations can be studied in terms of the fully-resolved behavioral phenotypes of individual cells.” (Line 116ff)

References

1. Hartmann, R., Jeckel, H., Jelli, E., Singh, P.K., Vaidya, S., Bayer, M., Vidakovic, L., Díaz-Pascual, F., Fong, J.C.N., Dragoš, A., Besharova, O., Nadell, C.D., Sourjik, V., Kovács, Á.T., Yildiz, F.H. & Drescher, K. BiofilmQ, a software tool for quantitative image analysis of microbial biofilm communities. *BioRxiv* 735423 (2019).
2. Hartmann, R., Singh, P.K., Pearce, P., Mok, R., Song, B., Díaz-Pascual, F., Dunkel, J. & Drescher, K. Emergence of three-dimensional order and structure in growing biofilms. *Nature Physics* **15**, 251-256 (2019).
3. Berg, S., Kutra, D., Kroeger, T., Straehle, C.N., Kausler, B.X., Haubold, C., Schiegg, M., Ales, J., Beier, T., Rudy, M., Eren, K., Cervantes, J.I., Xu, B., Beuttenmueller, F., Wolny, A., Zhang, C., Koethe, U., Hamprecht, F.A. & Kreshuk, A. ilastik: interactive machine learning for (bio)image analysis. *Nature Methods* (2019).
4. McQuin, C., Goodman, A., Chernyshev, V., Kametsky, L., Cimini, B.A., Karhohs, K.W., Doan, M., Ding, L., Rafelski, S.M., Thirstrup, D., Wiegraebe, W., Singh, S., Becker, T., Caicedo, J.C. & Carpenter, A.E. CellProfiler 3.0: Next-generation image processing for biology. *PLoS Biol* **16**, e2005970 (2018).
5. Weigert, M., Schmidt, U., Haase, R., Sugawara, K. & Myers, G. Star-convex Polyhedra for 3D Object Detection and Segmentation in Microscopy. in *2020 IEEE Winter Conference on Applications of Computer Vision (WACV)* 3655-3662 (2020).
6. Stringer, C., Wang, T., Michaelos, M. & Pachitariu, M. Cellpose: a generalist algorithm for cellular segmentation. *bioRxiv*, 2020.2002.2002.931238 (2020).
7. Jaccard, P. The distribution of the flora in the alpine zone. *New Phytologist* **11**, 37-50 (1912).
8. Liu, J., Prindle, A., Humphries, J., Gabalda-Sagarra, M., Asally, M., Lee, D.Y., Ly, S., Garcia-Ojalvo, J. & Suel, G.M. Metabolic co-dependence gives rise to collective oscillations within biofilms. *Nature* **523**, 550-554 (2015).
9. Prindle, A., Liu, J., Asally, M., Ly, S., Garcia-Ojalvo, J. & Suel, G.M. Ion channels enable electrical communication in bacterial communities. *Nature* **527**, 59-63 (2015).
10. Humphries, J., Xiong, L., Liu, J., Prindle, A., Yuan, F., Arjes, H.A., Tsimring, L. & Suel, G.M. Species-Independent Attraction to Biofilms through Electrical Signaling. *Cell* **168**, 200-209 e212 (2017).
11. Liu, J., Martinez-Corral, R., Prindle, A., Lee, D.-y.D., Larkin, J., Gabalda-Sagarra, M., Garcia-Ojalvo, J. & Suel, G.M. Coupling between distant biofilms and emergence of nutrient time-sharing. *Science* **356**, 638-642 (2017).

12. Mitri, S. & Foster, K.R. The genotypic view of social interactions in microbial communities. *Annu Rev Genet* **47**, 247-273 (2013).
13. Drescher, K., Nadell, C.D., Stone, H.A., Wingreen, N.S. & Bassler, B.L. Solutions to the public goods dilemma in bacterial biofilms. *Current biology : CB* **24**, 50-55 (2014).
14. Persat, A., Nadell, C.D., Kim, M.K., Ingremeau, F., Siryaporn, A., Drescher, K., Wingreen, N.S., Bassler, B.L., Gitai, Z. & Stone, H.A. The mechanical world of bacteria. *Cell* **161**, 988-997 (2015).
15. Papenfort, K. & Bassler, B.L. Quorum sensing signal-response systems in Gram-negative bacteria. *Nat Rev Microbiol* **14**, 576-588 (2016).
16. Nadell, C.D., Drescher, K. & Foster, K.R. Spatial structure, cooperation and competition in biofilms. *Nat Rev Microbiol* **14**, 589-600 (2016).
17. Kroos, L. Highly Signal-Responsive Gene Regulatory Network Governing Myxococcus Development. *Trends Genet* **33**, 3-15 (2017).
18. Moon, S., Yan, R., Kenny, S.J., Shyu, Y., Xiang, L., Li, W. & Xu, K. Spectrally resolved, functional super-resolution microscopy reveals nanoscale compositional heterogeneity in live-cell membranes. *Journal of the American Chemical Society* **139**, 10944-10947 (2017).
19. Bramkamp, M. & Lopez, D. Exploring the existence of lipid rafts in bacteria. *Microbiol. Mol. Biol. Rev.* **79**, 81-100 (2015).
20. Zou, S.B., Hersch, S.J., Roy, H., Wiggers, J.B., Leung, A.S., Buranyi, S., Xie, J.L., Dare, K., Ibba, M. & Navarre, W.W. Loss of elongation factor P disrupts bacterial outer membrane integrity. *Journal of bacteriology* **194**, 413-425 (2012).
21. Gonelimali, F.D., Lin, J., Miao, W., Xuan, J., Charles, F., Chen, M. & Hatab, S.R. Antimicrobial properties and mechanism of action of some plant extracts against food pathogens and spoilage microorganisms. *Frontiers in microbiology* **9**, 1639 (2018).
22. Parasassi, T., De Stasio, G., d'Ubaldo, A. & Gratton, E. Phase fluctuation in phospholipid membranes revealed by Laurdan fluorescence. *Biophysical journal* **57**, 1179-1186 (1990).
23. Strahl, H. & Hamoen, L.W. Membrane potential is important for bacterial cell division. *Proceedings of the National Academy of Sciences* **107**, 12281-12286 (2010).
24. Prindle, A., Liu, J., Asally, M., Ly, S., Garcia-Ojalvo, J. & Süel, G.M. Ion channels enable electrical communication in bacterial communities. *Nature* **527**, 59-63 (2015).

Response to Reviewer Comments (revised submission)

Reviewer #1 (Remarks to the Author):

The authors have addressed my concerns and suggestions with new analyses, which I appreciate seeing. As before, I find the approach well justified, well executed and successful, and I don't have any remaining caveats to add to these statements.

- Marius Pachitariu

Reviewer #2 (Remarks to the Author):

The authors have answered all my questions. I have no further comments.

- Thorsten Wohland

Reply: We again thank all the reviewer for their thoughtful and constructive comments on our manuscript.